# Circulation of pertussis and poor protection against diphtheria among middle-aged adults in 18 European countries

Guy Berbers[1✉], Pieter van Gageldonk [1], Jan van de Kassteele [1], Ursula Wiedermann[2], Isabelle Desombere[3], Tine Dalby [4], Julie Toubiana [5], Sotirios Tsiodras[6], Ildikó Paluska Ferencz[7], Kathryn Mullan[8], Algirdas Griskevicius[9], Tatjana Kolupajeva[10], Didrik Frimann Vestrheim[11], Paula Palminha[12], Odette Popovici[13], Lena Wehlin [14], Tamara Kastrin [15], Lucia Maďarová[16], Helen Campbell [17], Csaba Ködmön [18], Sabrina Bacci[18], Alex-Mikael Barkoff [19], Qiushui He [19,20✉] & the Serosurveillance Study Team*

Reported incidence of pertussis in the European Union (EU) and the European Economic Area (EEA) varies and may not reflect the real situation, while vaccine-induced protection against diphtheria and tetanus seems sufficient. We aimed to determine the seroprevalence of DTP antibodies in EU/EEA countries within the age groups of 40–49 and 50–59 years. Eighteen countries collected around 500 samples between 2015 and 2018 (N = 10,302) which were analysed for IgG-DTP specific antibodies. The proportion of sera with pertussis toxin antibody levels ≥100 IU/mL, indicative of recent exposure to pertussis was comparable for 13/18 countries, ranging between 2.7–5.8%. For diphtheria the proportion of sera lacking the protective level (<0.1 IU/mL) varied between 22.8–82.0%. For tetanus the protection was sufficient. Here, we report that the seroprevalence of pertussis in these age groups indicates circulation of *B. pertussis* across EU/EEA while the lack of vaccine-induced seroprotection against diphtheria is of concern and deserves further attention.

[1] National Institute of Public Health and the Environment, Bilthoven, The Netherlands. [2] Medical University of Vienna, Vienna, Austria. [3] Sciensano, Public Health Belgium, Brussels, Belgium. [4] Statens Serum Institut, Copenhagen, Denmark. [5] Institut Pasteur, Paris, France. [6] Hellenic Center for Disease Control, Athens, Greece. [7] National Center for Epidemiology, Budapest, Hungary. [8] National Virus reference laboratory, Dublin, Ireland. [9] National Public Health Surveillance Laboratory, Vilnius, Lithuania. [10] Molecular Biology Department, NRL, Riga, Latvia. [11] Norwegian Institute of Public Health, Oslo, Norway. [12] National Institute of Health, Lisbon, Portugal. [13] National Institute of Public Health, Bucharest, Romania. [14] Public Health Agency of Sweden, Stockholm, Sweden. [15] Slovenia National Laboratory of Health, Environment and Food, Ljubljana, Slovenia. [16] Regional Authority of Public Health, Banská Bystrica, Slovak Republic. [17] Public Health England, London, UK. [18] European Center for Disease Prevention and Control, Stockholm, Sweden. [19] University of Turku, Turku, Finland. [20] Capital Medical University, Beijing, China. *A list of authors and their affiliations appears at the end of the paper. ✉email: guy.berbers@rivm.nl; qiushui.he@utu.fi

W hooping cough or pertussis is a highly infectious disease that has resurged since the 1990s despite high vaccine coverage. The epidemic of 2012 in several countries resulted in the highest incidence of morbidity and mortality since the large-scale introduction of vaccines in the 1950s. The increase in the number of deaths in neonates, who are too young to be (fully) vaccinated, is particularly alarming. These newborns represent the highest-risk group for fatal pertussis disease, and often parents and close family members (e.g. grandparents) are the main source of infection[1,2].

The incidence of pertussis in European countries varies from 0.01 up to 50 per 100,000 inhabitants[3,4]. Moreover, most natural infections among adolescents and adults due to *Bordetella pertussis* result in mild or subclinical disease and are often not reported[5]. This underreporting of cases is recognised and the estimated rate of reported infections between European countries ranges from 1/1000 up to 33/1000 inhabitants, yearly[6,7]. To get a better estimate of the circulation of *B. pertussis* in the population, sero-epidemiology is a valuable tool complementary to clinical surveillance programmes. Serosurveillance of infections covered by the national vaccination programmes is important because it provides relevant information about the burden of infection and the immunological status of the population, and thus provides a tool to evaluate the risk of infection for not yet vaccinated infants. Furthermore, seroprevalence studies offer an opportunity to study waning immunity based on antibody decay in the population. Several European countries have performed such studies for pertussis (Belgium, Denmark, Italy, the Netherlands, Sweden and United Kingdom)[8–13]. However, these studies are based on antibody decay only and do not take into account other immunological parameters like cellular responses affecting the whole immunity against disease. Still, they reflect one side of the immunological protection induced by antibody seroprevalence. Furthermore, comparing serosurveillance studies between countries is not easy due to the wide variety of cohort selection criteria and laboratory tests that were used[9].

No internationally accepted correlate of protection for pertussis has been established. However, because pertussis toxin (PT) is specific for *B. pertussis*, the level of antibodies against PT (IgG-PT in IU/mL) is used as an aetiological marker of pertussis. Based on this marker the proportion of recent exposures in the population can be estimated provided that individuals vaccinated at least in the last 2 years are excluded[14,15].

In contrast to pertussis, the incidence of diphtheria and tetanus according to the data reported to ECDC has been very low in the last decade across EU/EEA countries, due to the longstanding vaccination programmes and high coverage, indicating that these vaccines seem to confer better protection than the pertussis vaccines[16]. Almost all EU/EEA countries reported a coverage of >90% for the third infant dose of DTP in the last decade[17]. Diphtheria re-emerges when vaccination programmes are compromised and outbreaks have been observed following drops in vaccination coverage to 60 and 80% in the former Soviet Union and former Soviet republics during 1990–1995 (ref. [18]), and more recently in Venezuela[19].

The aim of this study was to determine the seroprevalence levels of IgG-PT, IgG-Dt and IgG-TT within two specific age groups (i.e. 40–49 and 50–59 years of age) in EU/EEA countries to explore the proportion of sera indicative for a recent exposure to pertussis to determine the circulation of *B. pertussis*, and the persistence of vaccine-induced protection against diphtheria and tetanus in EU/EEA.

## Results

**Study characteristics**. From the 28 invited countries, 18 countries agreed to participate and shipped around 250 samples for both targeted age cohorts of 40–49 and 50–59 years to the RIVM. The characteristics of the sample collections like collection period (2015–2018) and locations are summarised in Table 1. The sex distribution of the samples was obtained from 16 countries. To avoid a selection bias, all samples (N = 1644) from the two age groups collected in a national serosurveillance study in the Netherlands were included. UK data comprised samples from England only. Altogether, the number of subjects included was 10,302.

**Antibody prevalence for pertussis**. The total percentages of sera per country with a level for IgG-PT ≥ 100 IU/mL varied between 0.0% (Finland) and 9.7% (Norway) with 13/18 countries showing a level between 2.7 and 5.8% (Fig. 1 and Table 2). The levels between 50 and 100 IU/mL for IgG-PT (Table 2) ranged between 4.8% (United Kingdom) and 9.9% (France) for all countries, excluding Norway with a level of 12.5%. The proportion of subjects with no detectable antibodies, IgG-PT < 0.85 IU/mL, varied between 2.0% (Norway) and 9.8% (Greece) (Table 2). The seroprevalence (IgG-PT ≥ 100 IU/mL) of the two age cohorts separately and in total per country is illustrated in Fig. 2 with a subdivision per sex. We found no influence of age and sex on the seroprevalence overall (p = 0.846 and p = 0.802 resp.), but the country effect was significant (p = 0.023, Supplementary Table 1). For overall seroprevalence of IgG-PT levels 50–100 IU/mL, also no effect was found for age and sex (p = 0.212, p = 0.082 resp.) and the country effect remained significant (p = 0.007), while for the IgG-PT level ≥50 IU/mL all three categories were significant (p = 0.038, p = 0.020, p < 0.001, respectively, Supplementary Table 1). The geometric mean concentration (GMC) values for IgG-PT antibodies ranged from 7.2 to 14.8 IU/mL within the 18 countries (Table 3). Significant GMC differences between the age cohorts were found for Austria, Lithuania, Latvia, the Netherlands and Norway with elevated IgG-PT concentrations in the 50–59 year cohort, except for Austria (Table 3). Between sex, significant GMC differences could be observed in the total cohort for Finland, Hungary, the Netherlands, Portugal and UK with elevated IgG-PT concentrations in males (Supplementary Fig. 1).

**Antibody prevalence for diphtheria**. The proportion of sera with Dt antibody levels below the basic immunity level of 0.01 IU/mL varied between 4% (Finland) and 43% (Greece) and for the protective level of 0.1 IU/mL from 23% for Finland up to around 80% for Greece, Ireland, Romania and United Kingdom (Fig. 1 and Table 2). Age, sex and country had a significant influence on the seroprotection for both cut-offs (p < 0.001, Supplementary Table 1). Significant differences for sex within the countries were found for the levels <0.01 IU/mL in the 40–49 years groups for Denmark, the Netherlands and Sweden, in the 50–59 year olds for Hungary, the Netherlands, Slovak Republic and United Kingdom and in the total cohorts for Denmark, Hungary, the Netherlands, Slovak Republic and United Kingdom (Fig. 3). For the levels <0.1 IU/mL differences in sex were found in the 40–49 year olds for Denmark, France, Hungary, Ireland and the Netherlands, in the 50–59 year olds for Denmark and the Netherlands, and in the total cohorts for Denmark, France, Ireland and the Netherlands. The GMCs of IgG-Dt levels were low for all participating countries, not exceeding 0.1 IU/mL in the total cohorts for 11/18 countries (Table 3). The GMCs in the 50–59 year olds were always lower than those of the 40–49 year olds, except for the Netherlands and Romania equally low and a significant difference in GMC between the age groups was found in 11/18 countries (Table 3).

**Table 1 Summary of sample collections: period of sample collection, number of samples collected by age group, type of samples collected and geographical origin by country.**

| Country | Collection time | Number of samples | | | Type of samples | Geographical origin |
|---|---|---|---|---|---|---|
| | | 40–49 y | 50–59 y | Total | | |
| Austria (AT) | 2015–2016 | 250 | 250 | 500 | Leftover samples | Vienna, Lower Austria, Carinthia, Salzburg, Burgenland |
| Belgium (BE) | 2017 | 252 | 252 | 504 | Leftover samples | Liège, Charleroi, Brussels, Bruges, Ghent |
| Denmark (DK) | 2015–2016 | 249 | 242 | 491 | Leftover samples (diagnostic laboratories) | Whole country |
| Finland (FI) | 2015–2016 | 250 | 250 | 500 | Leftover samples (patients with suspected coeliac disease) | Varsinais-Suomi, Pohjanmaa |
| France (FR) | 2015–2016 | 299 | 298 | 597 | Leftover samples | Institut Pasteur, Paris |
| Greece (GR) | 2015 | 250 | 250 | 501 | Leftover samples | Attikon University General Hospital, Athens |
| Hungary (HU) | 2017 | 260 | 273 | 533 | Leftover samples (health care workers) | Békés, Budapest, Csongrád, Hajdú, Szabolcs |
| Ireland (IE) | 2016–2017 | 250 | 249 | 499 | Leftover samples (patients) | Whole country |
| Latvia (LV) | 2015–2016 | 250 | 250 | 500 | Leftover samples | Riga |
| Lithuania (LT) | 2016 | 250 | 250 | 500 | Leftover samples (people admitted to outpatient clinics) | Kaunas |
| Netherlands (NL) | 2016–2017 | 830 | 814 | 1644 | Population-based seroprevalence study | 48 municipalities of the Netherlands |
| Norway (NO) | 2015–2016 | 251 | 251 | 502 | Leftover samples (Immunol. and Clin. Biochemistry Dept.) | Akershus |
| Portugal (PT) | 2015–2016 | 250 | 250 | 500 | Population-based seroprevalence study | Alentejo, Algarve, Centro, Lisboa, Norte, Açores, Madeira |
| Romania (RO) | 2018 | 252 | 252 | 504 | Leftover samples (emergency hospital laboratories) | 41 counties of Romania+Bucuresti Municipality |
| Slovak Republic (SK) | 2016–2018 | 250 | 250 | 500 | Leftover samples (patients inspected for borreliosis) | Banská Bystrica |
| Slovenia (SI) | 2015–2016 | 263 | 261 | 524 | Leftover samples (diagnostic borreliosis and measles) | Kranj, Maribor, Nova Gorica, Ljubljana |
| Sweden (SE) | 2016 | 253 | 251 | 504 | Leftover samples (clinical chemistry labs) | Halland, Jämtland/Härjedalen, Jönköping, Kalmar, Skåne, Stockholm, Västerbotten, Västra, Götaland, Örebro, Östergötland |
| United Kingdom (UK) | 2015–2016 | 250 | 249 | 499 | Leftover samples (genitourinary medicine) | Manchester, Exeter, Newcastle, Leicester, Leeds, London, Cambridge |

The participating countries are listed in alphabetical order of the complete name and not according to the abbreviation.

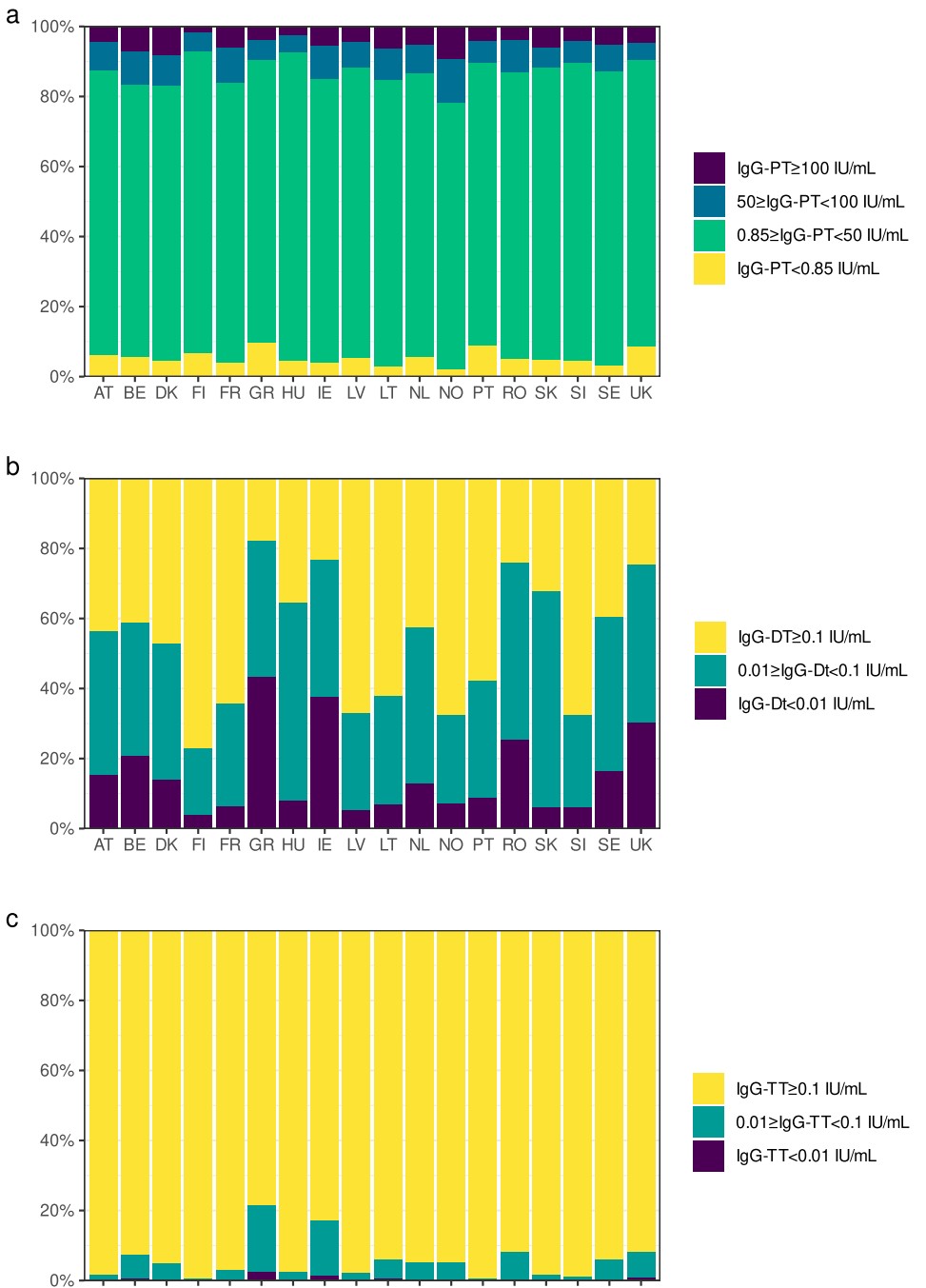

**Fig. 1 Relative distribution of samples by IgG intervals and by country.** IgG-PT (**a**), IgG-Dt (**b**) and IgG-TT (**c**). Proportion of pertussis seroprevalence divided in samples <0.85 IU/mL (yellow), 0.85 to <50 IU/mL (green), 50 to <100 IU/mL (blue) and ≥100 IU/mL (purple) and seroprotection against diphtheria and tetanus in <0.01 IU/mL (purple), 0.01 to <0.1 IU/mL (green) and ≥0.1 IU/mL (yellow) within the 18 countries. The bars sum up to 100%. Abbreviations of all participating countries and the number of samples included in the study are listed in Table 1.

**Antibody prevalence for tetanus**. In contrast with diphtheria, the seroprotection levels for tetanus were sufficient with only very few sera lacking basic immunity (Fig. 1 and Table 2). The proportion of sera with levels below 0.01 IU/mL ranged from 0 to 1.2%, apart from Greece (2.4%). For the total cohort, seven countries were considered as fully protected (Austria, Finland, France, Hungary, Latvia, Portugal and Slovenia). The protective level of 0.1 IU/mL was reached in more than 90% of the sera in all countries, apart from Greece (79%) and Ireland (83%). In the other 16 countries the proportion of sera with unprotected levels (<0.1 IU/mL)

ranged from 0.4 to 8.2%. Whereas sex and country had significant impact on the seroprotection level (<0.1 IU/mL, $p \leq 0.001$), no significant impact at the unprotected level (<0.01 IU/mL) was found for age, sex and country ($p = 0.902$, $p = 0.986$) $p = 0.491$, resp., Supplementary Table 1). Between countries no significant sex differences were found for the unprotected level in both age groups and the total cohort (Fig. 4). However, for the seroprotection level sex differences were found in the age group 40–49 years for Sweden, in the 50–59 years for Denmark, Hungary, Ireland and Latvia, and in the total cohorts for Belgium,

**Table 2 Percentages of pertussis seroprevalence and Dt and TT seroprotection per country and by age group.**

| Country | Age group | N total | IgG-PT (IU/mL) | | | | | | IgG-Dt (IU/mL) | | | | IgG-TT (IU/mL) | | | |
|---|---|---|---|---|---|---|---|---|---|---|---|---|---|---|---|---|
| | | | <LLOQ | 50 to <100 | | ≥100 | | | <0.01 | | <0.1 | | <0.01 | | <0.1 | |
| | | | % | % | N | % | a% corr. | N | % | N | % | N | % | N | % | N |
| AT | 40–49 | 250 | | 11.2 | 28 | 5.2 | 4.3 | 13 | 10.4 | 26 | 53.6 | 134 | 0.0 | 0 | 1.6 | 4 |
| | 50–59 | 250 | | 5.2 | 13 | 3.6 | 2.2 | 9 | 20.0 | 50 | 59.2 | 148 | 0.0 | 0 | 1.6 | 4 |
| | Total | 500 | 6.0 | 8.2 | 41 | 4.4 | 3.2 | 22 | 15.2 | 76 | 56.4 | 282 | 0.0 | 0 | 1.6 | 8 |
| BE | 40–49 | 252 | | 7.1 | 18 | 7.5 | 7.4 | 19 | 13.5 | 34 | 54.8 | 138 | 0.0 | 0 | 6.0 | 15 |
| | 50–59 | 252 | | 11.5 | 29 | 7.1 | 6.8 | 18 | 27.8 | 70 | 63.1 | 159 | 1.2 | 3 | 8.7 | 22 |
| | Total | 504 | 5.6 | 9.3 | 47 | 7.3 | 7.1 | 37 | 20.6 | 104 | 58.9 | 297 | 0.6 | 3 | 7.3 | 37 |
| DK | 40–49 | 249 | | 7.6 | 19 | 5.6 | 4.9 | 14 | 12.9 | 32 | 52.6 | 131 | 0.4 | 1 | 4.0 | 10 |
| | 50–59 | 242 | | 9.5 | 23 | 11.2 | 12.1 | 27 | 15.3 | 37 | 52.9 | 128 | 0.0 | 0 | 5.8 | 14 |
| | Total | 491 | 4.5 | 8.6 | 42 | 8.4 | 8.4 | 41 | 14.1 | 69 | 52.8 | 259 | 0.2 | 1 | 4.9 | 24 |
| FI | 40–49 | 250 | | 4.0 | 10 | 2.0 | 0.1 | 5 | 3.2 | 8 | 20.4 | 51 | 0.0 | 0 | 0.4 | 1 |
| | 50–59 | 250 | | 6.8 | 17 | 1.6 | b0.0 | 4 | 4.4 | 11 | 25.2 | 63 | 0.0 | 0 | 0.4 | 1 |
| | Total | 500 | 6.6 | 5.4 | 27 | 1.8 | b0.0 | 9 | 3.8 | 19 | 22.8 | 114 | 0.0 | 0 | 0.4 | 2 |
| FR | 40–49 | 299 | | 9.4 | 28 | 4.4 | 3.2 | 13 | 2.0 | 6 | 23.1 | 69 | 0.0 | 0 | 2.7 | 8 |
| | 50–59 | 298 | | 10.4 | 31 | 8.1 | 8.0 | 24 | 10.7 | 32 | 48.0 | 143 | 0.0 | 0 | 3.4 | 10 |
| | Total | 597 | 4.0 | 9.9 | 59 | 6.2 | 5.6 | 37 | 6.4 | 38 | 35.5 | 212 | 0.0 | 0 | 3.0 | 18 |
| GR | 40–49 | 250 | | 6.0 | 15 | 4.8 | 3.8 | 12 | 26.8 | 67 | 72.4 | 181 | 0.0 | 0 | 11.6 | 29 |
| | 50–59 | 250 | | 5.2 | 13 | 3.2 | 1.7 | 8 | 59.6 | 149 | 91.6 | 229 | 4.8 | 12 | 31.2 | 78 |
| | Total | 501 | 9.8 | 5.6 | 28 | 4.0 | 2.7 | 20 | 43.3 | 217 | 82.0 | 411 | 2.4 | 12 | 21.4 | 107 |
| HU | 40–49 | 260 | | 5.0 | 13 | 1.5 | b0.0 | 4 | 3.5 | 9 | 60.4 | 157 | 0.0 | 0 | 2.7 | 7 |
| | 50–59 | 273 | | 4.8 | 13 | 3.7 | 2.3 | 10 | 12.5 | 34 | 68.1 | 186 | 0.0 | 0 | 2.2 | 6 |
| | Total | 533 | 4.5 | 4.9 | 26 | 2.6 | 0.9 | 14 | 8.1 | 43 | 64.4 | 343 | 0.0 | 0 | 2.4 | 13 |
| IE | 40–49 | 250 | | 8.4 | 21 | 5.6 | 4.8 | 14 | 33.2 | 83 | 76.4 | 191 | 0.4 | 1 | 12.8 | 32 |
| | 50–59 | 249 | | 10.4 | 26 | 5.6 | 4.8 | 14 | 41.8 | 104 | 77.1 | 192 | 2.0 | 5 | 21.3 | 53 |
| | Total | 499 | 3.8 | 9.4 | 47 | 5.6 | 4.8 | 28 | 37.5 | 187 | 76.8 | 383 | 1.2 | 6 | 17.0 | 85 |
| LV | 40–49 | 250 | | 4.8 | 12 | 4.0 | 2.7 | 10 | 1.2 | 3 | 22.0 | 55 | 0.0 | 0 | 0.8 | 2 |
| | 50–59 | 250 | | 9.6 | 24 | 5.2 | 4.3 | 13 | 9.2 | 23 | 44.0 | 110 | 0.0 | 0 | 3.6 | 9 |
| | Total | 500 | 5.2 | 7.2 | 36 | 4.6 | 3.5 | 23 | 5.2 | 26 | 33.0 | 165 | 0.0 | 0 | 2.2 | 11 |
| LT | 40–49 | 250 | | 7.2 | 18 | 4.8 | 3.8 | 12 | 2.8 | 7 | 27.6 | 69 | 0.4 | 1 | 3.6 | 9 |
| | 50–59 | 250 | | 10.8 | 27 | 8.0 | 8.0 | 20 | 10.8 | 27 | 48.4 | 121 | 0.8 | 2 | 8.4 | 21 |
| | Total | 500 | 2.8 | 9.0 | 45 | 6.4 | 5.8 | 32 | 6.8 | 34 | 38.0 | 190 | 0.6 | 3 | 6.0 | 30 |
| NL | 40–49 | 830 | | 8.6 | 71 | 4.9 | 3.9 | 41 | 12.3 | 102 | 56.5 | 469 | 0.5 | 4 | 4.6 | 38 |
| | 50–59 | 814 | | 7.5 | 61 | 5.9 | 5.2 | 48 | 13.3 | 108 | 58.6 | 477 | 0.1 | 1 | 5.8 | 47 |
| | Total | 1644 | 5.6 | 8.0 | 132 | 5.4 | 4.5 | 89 | 12.8 | 210 | 57.5 | 946 | 0.3 | 5 | 5.2 | 85 |
| NO | 40–49 | 251 | | 12.0 | 30 | 7.6 | 7.4 | 19 | 5.6 | 14 | 31.9 | 80 | 0.0 | 0 | 3.6 | 9 |
| | 50–59 | 251 | | 13.1 | 33 | 11.2 | 12.1 | 28 | 8.8 | 22 | 32.7 | 82 | 0.4 | 1 | 6.8 | 17 |
| | Total | 502 | 2.0 | 12.5 | 63 | 9.4 | 9.7 | 47 | 7.2 | 36 | 32.3 | 162 | 0.2 | 1 | 5.2 | 26 |
| PT | 40–49 | 250 | | 6.0 | 15 | 4.0 | 2.7 | 10 | 7.6 | 19 | 42.4 | 106 | 0.0 | 0 | 0.4 | 1 |
| | 50–59 | 250 | | 6.4 | 16 | 4.4 | 3.2 | 11 | 10.0 | 25 | 41.6 | 104 | 0.0 | 0 | 0.4 | 1 |
| | Total | 500 | 8.8 | 6.2 | 31 | 4.2 | 2.9 | 21 | 8.8 | 44 | 42.0 | 210 | 0.0 | 0 | 0.4 | 2 |
| RO | 40–49 | 252 | | 7.1 | 18 | 4.0 | 2.7 | 10 | 25.8 | 65 | 77.4 | 195 | 0.0 | 0 | 4.0 | 10 |
| | 50–59 | 252 | | 11.1 | 28 | 4.0 | 2.7 | 10 | 25.0 | 63 | 74.2 | 187 | 0.4 | 1 | 12.3 | 31 |
| | Total | 504 | 5.2 | 9.1 | 46 | 4.0 | 2.6 | 20 | 25.4 | 128 | 75.8 | 382 | 0.2 | 1 | 8.1 | 41 |
| SK | 40–49 | 250 | | 6.0 | 15 | 7.2 | 6.9 | 18 | 5.2 | 13 | 64.4 | 161 | 0.0 | 0 | 2.0 | 5 |
| | 50–59 | 250 | | 5.6 | 14 | 4.8 | 3.8 | 12 | 6.8 | 17 | 71.2 | 178 | 0.4 | 1 | 1.2 | 3 |
| | Total | 500 | 4.8 | 5.8 | 29 | 6.0 | 5.3 | 30 | 6.0 | 30 | 67.8 | 339 | 0.2 | 1 | 1.6 | 8 |
| SI | 40–49 | 263 | | 7.2 | 19 | 4.6 | 3.5 | 12 | 1.9 | 5 | 26.6 | 70 | 0.0 | 0 | 0.4 | 1 |
| | 50–59 | 261 | | 5.0 | 13 | 3.8 | 2.5 | 10 | 10.3 | 27 | 37.9 | 99 | 0.0 | 0 | 1.5 | 4 |
| | Total | 524 | 4.4 | 6.1 | 32 | 4.2 | 2.9 | 22 | 6.1 | 32 | 32.3 | 169 | 0.0 | 0 | 1.0 | 5 |
| SE | 40–49 | 253 | | 9.5 | 24 | 6.3 | 5.8 | 16 | 11.5 | 29 | 58.5 | 148 | 0.0 | 0 | 5.5 | 14 |
| | 50–59 | 251 | | 6.0 | 15 | 4.0 | 2.7 | 10 | 21.1 | 53 | 62.2 | 156 | 0.4 | 1 | 6.4 | 16 |
| | Total | 504 | 3.2 | 7.7 | 39 | 5.2 | 4.2 | 26 | 16.3 | 82 | 60.3 | 304 | 0.2 | 1 | 6.0 | 30 |
| UK | 40–49 | 250 | | 4.0 | 10 | 5.2 | 4.3 | 13 | 24.8 | 62 | 71.6 | 179 | 1.2 | 3 | 7.6 | 19 |
| | 50–59 | 249 | | 5.6 | 14 | 4.4 | 3.3 | 11 | 35.7 | 89 | 79.1 | 197 | 0.4 | 1 | 8.8 | 22 |
| | Total | 499 | 8.6 | 4.8 | 24 | 4.8 | 3.7 | 24 | 30.3 | 151 | 75.4 | 376 | 0.8 | 4 | 8.2 | 41 |

aIgG anti-PT percentage (%) ≥100 IU/mL corrected for assay sensitivity (Se = 78%) and specificity (Sp = 98%).
bCalculated negative IgG anti-PT percentage (%) ≥100 IU/mL changed to 0.0%.
Abbreviations of all participating countries are listed in Table 1.

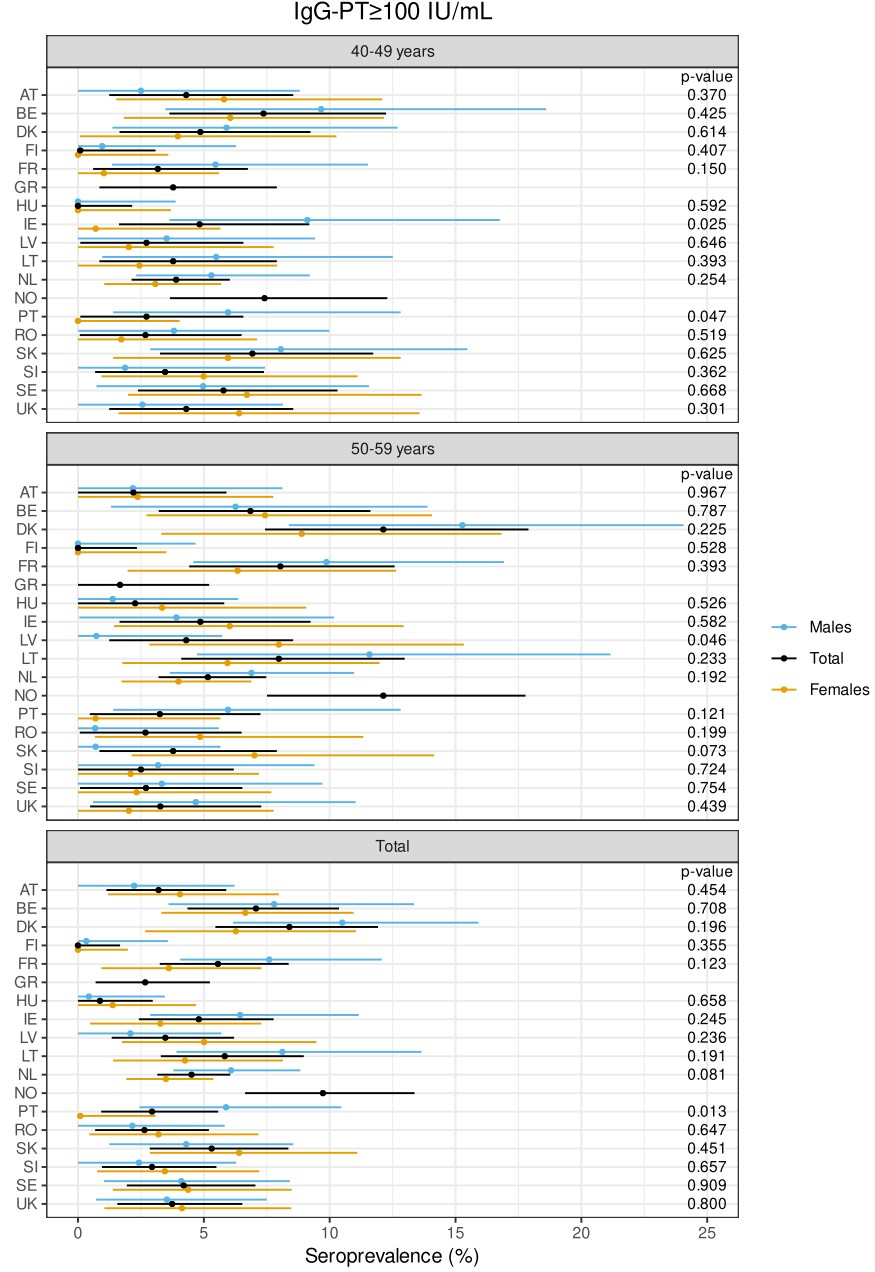

**Fig. 2 Percentage of pertussis infected sera in the two age groups (40–49 and 50–59) separately and in the total cohort, and subdivided by sex per country (Y-axis).** On the X-axis the percentage of seroprevalence for PT ≥ 100 IU/mL is displayed. The dots indicate the estimated seroprevalence, the bars the corresponding 95% confidence intervals. The estimates and p values of the differences are obtained by a binomial generalised linear regression model, in which a modified logit link function is used to correct for a specificity of 0.98 and a sensitivity of 0.78. Abbreviations of all participating countries and the number of samples included in the study are listed in Table 1.

Denmark, Hungary, Ireland, Latvia and Sweden. The GMCs of IgG-TT were above 1.0 IU/mL in 14 countries ranging from 1.15 to 3.16 IU/mL, and below 1.0 IU/mL in four countries ranging from 0.36 to 0.83 IU/mL. In 13 countries the GMCs in the 50–59 years olds were lower than in the 40–49 year olds. Significant differences in GMC between the age groups were found in France, Greece, Latvia and Romania (Table 3).

**Discussion**

The proportion of sera with an IgG-PT antibody level ≥100 IU/mL indicative for a recent exposure to pertussis was comparable for 13 out of 18 EU/EEA countries ranging between 2.7 and 5.8% with outliers up to 0.0 and 9.7% as illustrated by RCDCs (Fig. 5).

In addition, the GMCs of IgG-PT antibodies in all countries varied between 7 and 15 IU/mL, suggesting that the epidemiological situation for pertussis across EU/EEA is broadly similar. In contrast, for diphtheria the proportion of sera with no basic immunity showed a broad range between 3.8 and 43.3%. For the protective level these proportions ranged from 22.8% to 82.0%, suggesting that the protection against diphtheria is insufficient in older age cohorts in most EU/EEA countries. For tetanus the protection seems sufficient with only very few sera lacking basic immunity. More than 90% of the sera from all countries possessed protective levels except one country with 83%. To our best knowledge, this is the largest seroprevalence study of pertussis, diphtheria and tetanus conducted in EU/EEA since DTP vaccines

**Table 3 GMCs for IgG-PT, IgG-Dt and IgG-TT in IU/mL per country and by age group.**

| Country | Age group | IgG-PT (IU/mL) | | IgG-Dt (IU/mL) | | IgG-TT (IU/mL) | |
| --- | --- | --- | --- | --- | --- | --- | --- |
| | | GMC (95% CI) | *p* value | GMC (95% CI) | *p* value | GMC (95% CI) | *p* value |
| AT | 40–49 | 11.8 (9.8–14.1) | | 0.08 (0.06–0.10) | | 1.84 (1.64–2.07) | |
| | 50–59 | 7.8 (6.5–9.4) | | 0.05 (0.04–0.06) | | 1.85 (1.63–2.10) | |
| | Total | 9.6 (8.4–10.9) | 0.002 | 0.06 (0.06–0.07) | 0.008 | 1.85 (1.70–2.01) | 0.952 |
| BE | 40–49 | 12.1 (10.1–14.4) | | 0.07 (0.06–0.09) | | 1.64 (1.38–1.95) | |
| | 50–59 | 12.4 (10.4–14.9) | | 0.04 (0.03–0.05) | | 1.34 (1.09–1.64) | |
| | Total | 12.2 (10.8–13.9) | 0.829 | 0.05 (0.05–0.06) | <0.001 | 1.48 (1.30–1.69) | 0.136 |
| DK | 40–49 | 10.8 (9.0–13.0) | | 0.08 (0.07–0.10) | | 1.80 (1.52–2.13) | |
| | 50–59 | 13.3 (11.0–15.9) | | 0.08 (0.06–0.09) | | 2.02 (1.70–2.41) | |
| | Total | 12.0 (10.5–13.6) | 0.128 | 0.08 (0.07–0.09) | 0.739 | 1.91 (1.69–2.15) | 0.346 |
| FI | 40–49 | 7.4 (6.2–8.9) | | 0.29 (0.24–0.36) | | 3.23 (2.93–3.57) | |
| | 50–59 | 8.0 (6.7–9.6) | | 0.22 (0.18–0.27) | | 3.10 (2.77–3.47) | |
| | Total | 7.7 (6.8–8.8) | 0.544 | 0.25 (0.22–0.29) | 0.053 | 3.16 (2.94–3.41) | 0.582 |
| FR | 40–49 | 12.9 (10.9–15.2) | | 0.27 (0.23–0.33) | | 1.94 (1.72–2.20) | |
| | 50–59 | 14.0 (11.8–16.5) | | 0.09 (0.08–0.11) | | 1.28 (1.12–1.45) | |
| | Total | 13.4 (11.9–15.1) | 0.502 | 0.16 (0.14–0.18) | <0.001 | 1.58 (1.44–1.73) | <0.001 |
| GR | 40–49 | 8.0 (6.7–9.6) | | 0.03 (0.02–0.04) | | 0.62 (0.51–0.75) | |
| | 50–59 | 7.6 (6.3–9.1) | | 0.01 (0.01–0.01) | | 0.21 (0.17–0.26) | |
| | Total | 7.8 (6.9–8.9) | 0.675 | 0.01 (0.01–0.02) | <0.001 | 0.36 (0.31–0.42) | <0.001 |
| HU | 40–49 | 7.5 (6.3–8.9) | | 0.08 (0.07–0.10) | | 2.38 (2.02–2.79) | |
| | 50–59 | 8.2 (6.9–9.7) | | 0.05 (0.04–0.07) | | 2.24 (1.92–2.62) | |
| | Total | 7.8 (6.9–8.8) | 0.495 | 0.07 (0.06–0.08) | 0.006 | 2.31 (2.06–2.58) | 0.614 |
| IE | 40–49 | 10.7 (8.9–12.8) | | 0.02 (0.02–0.03) | | 0.79 (0.64–0.97) | |
| | 50–59 | 12.2 (10.1–14.6) | | 0.02 (0.02–0.02) | | 0.64 (0.50–0.81) | |
| | Total | 11.4 (10.0–12.9) | 0.317 | 0.02 (0.02–0.02) | 0.006 | 0.71 (0.60–0.83) | 0.184 |
| LV | 40–49 | 7.0 (5.8–8.4) | | 0.26 (0.21–0.31) | | 1.94 (1.70–2.20) | |
| | 50–59 | 11.1 (9.2–13.2) | | 0.11 (0.09–0.14) | | 1.42 (1.21–1.65) | |
| | Total | 8.8 (7.7–10.0) | <0.001 | 0.17 (0.15–0.20) | <0.001 | 1.66 (1.50–1.83) | 0.002 |
| LT | 40–49 | 10.6 (8.8–12.7) | | 0.17 (0.14–0.21) | | 1.29 (1.10–1.51) | |
| | 50–59 | 15.0 (12.5–18.0) | | 0.08 (0.07–0.10) | | 1.03 (0.85–1.24) | |
| | Total | 12.6 (11.1–14.3) | 0.007 | 0.12 (0.10–0.13) | <0.001 | 1.15 (1.02–1.30) | 0.076 |
| NL | 40–49 | 9.7 (8.7–10.7) | | 0.06 (0.06–0.07) | | 1.24 (1.13–1.36) | |
| | 50–59 | 11.2 (10.2–12.4) | | 0.06 (0.06–0.07) | | 1.20 (1.10–1.32) | |
| | Total | 10.4 (9.7–11.2) | 0.037 | 0.06 (0.06–0.07) | 0.700 | 1.22 (1.14–1.30) | 0.690 |
| NO | 40–49 | 13.0 (10.9–15.6) | | 0.17 (0.14–0.21) | | 1.45 (1.24–1.68) | |
| | 50–59 | 16.9 (14.1–20.2) | | 0.16 (0.13–0.19) | | 1.18 (0.98–1.41) | |
| | Total | 14.8 (13.1–16.9) | 0.047 | 0.16 (0.14–0.19) | 0.553 | 1.31 (1.16–1.47) | 0.084 |
| PT | 40–49 | 6.9 (5.7–8.2) | | 0.12 (0.10–0.15) | | 2.63 (2.36–2.92) | |
| | 50–59 | 8.8 (7.4–10.6) | | 0.11 (0.09–0.14) | | 2.63 (2.38–2.91) | |
| | Total | 7.8 (6.8–8.8) | 0.055 | 0.12 (0.10–0.14) | 0.688 | 2.63 (2.44–2.83) | 0.974 |
| RO | 40–49 | 9.9 (8.3–11.9) | | 0.03 (0.02–0.04) | | 0.80 (0.69–0.93) | |
| | 50–59 | 10.9 (9.1–13.0) | | 0.03 (0.02–0.04) | | 0.56 (0.47–0.66) | |
| | Total | 10.4 (9.1–11.8) | 0.479 | 0.03 (0.03–0.03) | 0.762 | 0.67 (0.60–0.75) | 0.002 |
| SK | 40–49 | 9.5 (7.9–11.4) | | 0.06 (0.05–0.08) | | 1.56 (1.36–1.80) | |
| | 50–59 | 8.2 (6.8–9.8) | | 0.05 (0.04–0.06) | | 1.64 (1.42–1.89) | |
| | Total | 8.8 (7.8–10.0) | 0.254 | 0.06 (0.05–0.07) | 0.168 | 1.60 (1.45–1.77) | 0.632 |
| SI | 40–49 | 10.4 (8.7–12.4) | | 0.18 (0.14–0.22) | | 2.03 (1.79–2.30) | |
| | 50–59 | 8.6 (7.2–10.3) | | 0.12 (0.09–0.14) | | 2.01 (1.76–2.30) | |
| | Total | 9.4 (8.3–10.7) | 0.150 | 0.14 (0.12–0.17) | 0.004 | 2.02 (1.85–2.21) | 0.916 |
| SE | 40–49 | 11.6 (9.7–13.9) | | 0.07 (0.06–0.09) | | 1.14 (0.97–1.36) | |
| | 50–59 | 9.4 (7.8–11.2) | | 0.05 (0.04–0.06) | | 1.22 (1.01–1.46) | |
| | Total | 10.4 (9.2–11.8) | 0.100 | 0.06 (0.05–0.07) | 0.012 | 1.18 (1.04–1.34) | 0.626 |
| UK | 40–49 | 7.3 (6.1–8.8) | | 0.03 (0.03–0.04) | | 0.86 (0.72–1.02) | |
| | 50–59 | 7.2 (6.0–8.6) | | 0.02 (0.02–0.03) | | 0.80 (0.67–0.95) | |
| | Total | 7.2 (6.4–8.2) | 0.901 | 0.03 (0.02–0.03) | 0.001 | 0.83 (0.73–0.93) | 0.578 |

*P* values are given for the difference between the two age groups. Abbreviations of all participating countries and the number of samples included in the study are listed in Table 1.

were introduced, using centralised testing of specific antibody levels to minimise the variation of methods used, and which enables direct comparison between all participating countries.

In Europe, vaccination programmes including whole-cell pertussis (wP) vaccines were implemented during the 1950s, so the majority of participants of this study would have received a wP vaccine. However, it may be expected that due to waning immunity the vaccinated participants are susceptible to infection just like the non-vaccinated and have been re-infected potentially with milder symptoms. From the late 1990s up until 2006 all European countries (except Poland) switched to acellular pertussis vaccines (aP). However, despite continuous high pertussis infant vaccination coverage in most countries (≥95%) the pathogen is still circulating. Based on the ECDC and WHO

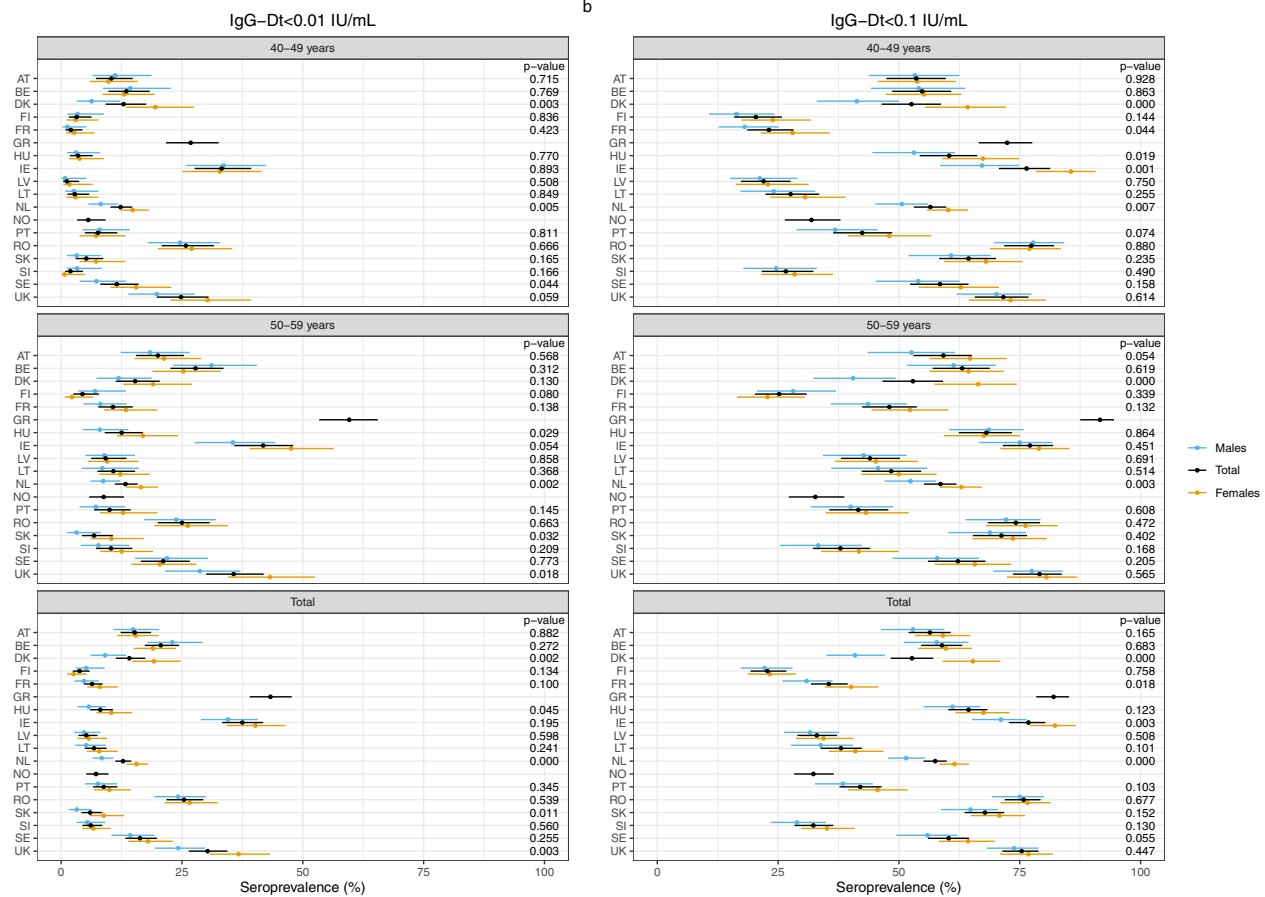

**Fig. 3 Percentage of sera protected against diphtheria in the two age groups (40–49 and 50–59) separately and in the total cohort, and subdivided by sex per country (Y-axis).** On the X-axis the percentage of lack of seroprotection against diphtheria for IgG-Dt <0.01 IU/mL (**a**) and <0.1 IU/mL (**b**) is displayed. The dots indicate the estimated seroprevalence; the bars the corresponding 95% confidence intervals. The estimates and p values of the differences are obtained by a binomial generalised linear regression model with logit link function. Abbreviations of all participating countries and the number of samples included in the study are listed in Table 1.

websites[3,4] an increase in the incidence of pertussis has been reported during the last decade in Austria, Belgium, Denmark, Ireland, Latvia, the Netherlands, Norway, Slovak Republic, Slovenia, Sweden and United Kingdom, although increased awareness and improved laboratory diagnostics by serology and PCR also could have contributed. After the initial increase, the incidence has remained high in most countries, while in the other countries (Finland, France, Greece, Hungary, Lithuania, Portugal and Romania) low incidence was reported.

In 11 countries, the serum collection period coincided with the increase and subsequent higher incidence. In the other seven countries, low incidence numbers were reported during that period. Distribution of the collection period into three groups (2015–2016, 2016–2017 and ≥2017) revealed no statistically significant influence on the pertussis seroprevalence data. However, as in many European countries booster vaccinations were implemented[20], this has most likely affected the circulation of the disease as described by the study of de Cellès et al. based on mathematical modelling showing the impact of childhood boosters to transmission of the disease[21]. Also, the geographical origin of the samples did not affect the pertussis serosurveillance outcome when the countries were divided into three groups of one location, 2–7 locations or whole country. The source of the serum samples was very diverse: from patients (three countries), from healthy people (three countries), but mostly of unknown origin consisting of leftover samples for diagnostics (12 countries). Therefore, the possibility that the different sources might have

affected the results seemed minimal considering the non-matching different outcomes per country and source. Whereas no age and sex effect on the seroprevalence results for IgG-PT ≥ 100 IU/mL for the whole study was observed, the country effect was very clear because the whole range of proportions of recently exposed participants in the EU/EEA was still quite large. This country effect might be explained by the differences in pertussis vaccination schedules, including adult boosters, and vaccines used in the EU/EEA countries throughout the years. Geography and density of the population did not seem to play a role, as Finland and Norway (extremes) are both low-density Nordic countries. A trend towards higher GMCs in males was observed in 14 countries reaching significance in four countries, while in the two other countries GMCs were almost identical between females and males. This might be due to booster vaccinations for the military service and/or a slightly better immune response upon natural infection in males. Sex-specific susceptibility for pertussis might be a relevant factor.

This cross-sectional seroprevalence study shows (low) circulation of pertussis among these middle-aged adults in EU/EEA despite well implemented childhood vaccination programmes and underscores the need for vigilant surveillance of pertussis. For only two countries (Finland and Hungary) the serosurveillance study is not sensitive enough and indicates to no pertussis circulation at all. Surprisingly, in these two countries pertussis case have been notified during 2015–2018 and specifically in Finland the reported number of pertussis during the study period

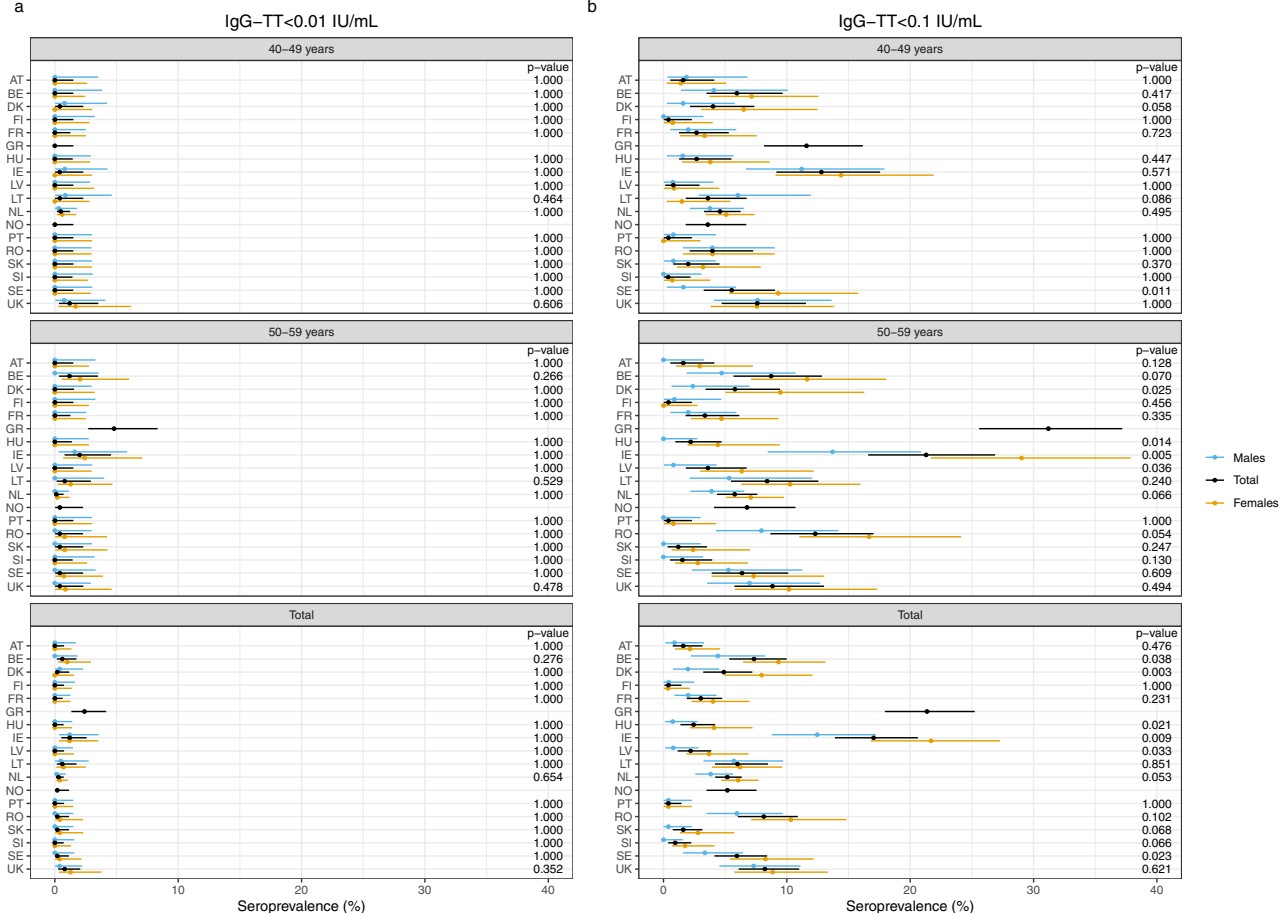

**Fig. 4 Percentage of sera protected against tetanus in the two age groups (40–49 and 50–59) separately and in the total cohort, and subdivided by sex per country (Y-axis).** On the *X*-axis the percentage of lack of seroprotection against tetanus for IgG-TT < 0.01 IU/mL (**a**) and <0.1 IU/mL (**b**) is displayed. The dots indicate the estimated seroprevalence, the bars the corresponding 95% confidence intervals. The estimates and *p* values of the differences are obtained by using exact binomial methods. Abbreviations of all participating countries and the number of samples included in the study are listed in Table 1.

(2015–2016) was 597 including 31 culture-proven cases, suggesting that *B. pertussis* is actually circulating in this country as well. This study also emphasises that a very large proportion of middle-aged adults in all 18 EU/EEA countries seem to have low to insufficient antibody levels against PT (<50 IU/mL) with a small proportion showing undetectable levels and therefore could be susceptible for pertussis infection as measured by antibodies. This situation is of concern for young infants at high risk for serious disease, because recent studies have shown that parents, siblings and close family (grandparents) are the main source of whooping cough in infants[1,2]. However, we need to stress that immunity against pertussis is not only based on anti-PT IgG antibodies. There are several other factors, e.g., cell-mediated immunity and antibodies against other pertussis antigens, contributing to protection against pertussis. But, the primary aim of this study was to detect circulation of pertussis in the adult population and antibodies are not generated unless there is an exposure to the pathogen or a recent vaccination against the disease. It should be mentioned that it is not possible to differentiate between a recent pertussis vaccination and a pertussis infection in the last 2 years but a recent pertussis vaccination is not very likely in these age cohorts (40–59 years) because such adult boosters are not included in the national immunisation programmes except for France and Austria and the coverage of these adult boosters is very low. The national vaccination programmes for pertussis are designed to protect vulnerable infants,

but despite the high vaccine coverage, an increasing incidence of pertussis among children has been reported. This has led to a renewed focus on how to protect infants, such as maternal immunisation, which might be needed to prevent pertussis but must be considered nationally and should take into account the local epidemiology[22]. In EU/EEA, ten countries have now implemented maternal immunisation which has been found highly effective at preventing pertussis in young infants from birth until they receive primary vaccinations[23,24]. Also, 11 countries have implemented more aP vaccinations, like adolescent boosters, boosters for military service, adult boosters and/or cocooning. However, only a high country-wide coverage of these boosters might have a significant influence on the pertussis incidence. Moreover, there is evidence that the immunogenicity of repeated aP boosters seems to diminish, although the persistence of aP vaccine-induced antibodies after a first booster in wP-primed adults appeared to be longer compared to children[25]. Furthermore, many studies have shown the kinetics of anti-PT IgG antibodies and how they quite rapidly wane after vaccination or even after infection, although more prolonged[26–28].

The reported pertussis cases of the countries participating in this study varied enormously ranging from 0.01 per 100,000 citizens up to 50/100,000, annually as illustrated in Supplementary Table 2 (refs. [3,4]). In this study we show a pertussis circulation in the adult population consistent with other serosurveillance studies[8–13], suggesting a large underreporting of adult pertussis

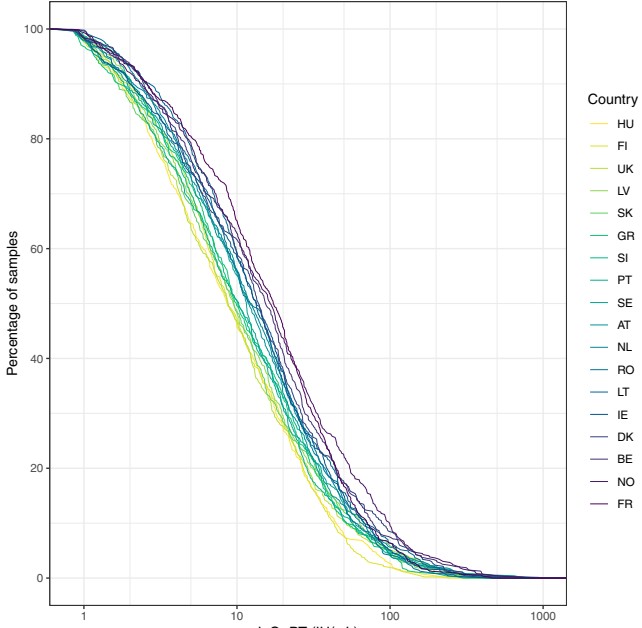

**Fig. 5 Reverse cumulative distribution curves for IgG-PT (IU/mL) illustrated by increasing values at 50% per country.** Values below the LLOQ are not included. Abbreviations of all participating countries and the number of samples included in the study are listed in Table 1.

cases. This is confirmed by calculating seroincidence rates from our cross-sectional IgG-PT antibody data[29,30]. We estimated rates to be between 0.02/person-year (Finland) and 0.05/person-year (Norway), corresponding to the ones given by Kretzschmar et al.[31]. Using 320 days as median seropositivity period with antibody levels IgG-PT ≥ 100 IU/mL[32] this corresponds to a seroprevalence of 1.8–4.4% in line with our serosurveillance results. Many of these cases will occur as mild infections with subclinical symptoms and are therefore not reported and not captured by routine surveillance systems[33]. Although the real ratio of pertussis infections to reported cases, and thus disease incidence, is unknown, it has been shown that between 13 and 25% of adults with prolonged cough have high levels of pertussis antibodies[34], demonstrating that the current monitoring system based on case reporting is under-ascertaining the true burden of disease. To get a better estimate of the circulation of *B. pertussis* in the population, (regular) sero-epidemiology is a valuable tool complementary to surveillance programmes based on case reporting. We endorse that serology is perhaps not the gold standard for laboratory diagnostic of pertussis but rather bacterial culture from nasopharyngeal swabs. However, the sensitivity of bacterial culture is relatively low, especially for adults. If we only rely on culture, the number of reported cases would be extremely low and we would not obtain a good picture of pertussis situation. Several studies have shown that pertussis vaccination conferred imperfect, but quite long-lasting protection and that pertussis aP/wP vaccinations directly decrease the number of pertussis cases and transmission[21,35–38]. Novel studies using mathematical models to estimate vaccine efficacy and protection especially in children showed that more than 65% of the children remained immune to pertussis after 5 years since the last booster dose and vaccine efficacy for a child booster is close to 75%, indicating good protection from the vaccine. Furthermore, boosting school children is more effective than boosting adult population as the contact rates among these two cohorts are different[21,35,36]. Although it is still clear that adding booster doses into national immunisation programmes (NIPs) will decrease the number of pertussis cases and

circulation of the disease, the effect of these booster vaccinations on the exposure of pertussis in our age groups seems still limited in our study. However, further investigations are clearly needed to conclude the real long-term protection of the current pertussis vaccinations.

The high proportion of sera with unprotected levels for diphtheria is of concern, leaving at least a quarter up to over three-quarters of the middle-aged adult population sampled not well protected against diphtheria. Since the infection is toxin mediated, protection will mainly be provided through specific antibodies, but after waning of these antibodies other cellular immunological mechanisms can still play a role in diphtheria immunity. Presumably the vaccine-induced antibody levels against diphtheria have waned in these middle-aged adult cohorts, while the original responses to the primary series in the first year of life would be expected to have been good and in many countries (several) boosters have been administered during childhood. In some countries DT boosters are administered when people travel to endemic countries, but this is too sporadic to influence the outcome of this study. From an epidemiological perspective the protection against diphtheria seems sufficient because no increase in cases has been noticed in EU/EEA during the last decades but import from locations with diphtheria outbreaks remains a real threat. Age, sex and country affected the seroprotection levels in this study. The age effect can be explained by waning immunity due to ageing. The country effect can be attributed to the different vaccination schedules, but not to the used vaccines because the Dt component is similar in most combination vaccines. The sex effect is rather surprising, and might be interpreted as a sex-specific difference in immunity but has been described recently in another study of the European diphtheria surveillance network[39]. Moreover, a trend towards higher IgG-Dt antibody levels in females compared to males in 12/16 countries was found reaching significance in five countries. Waning of diphtheria immunity over the years appears to proceed faster in males than in females. Overall, the protection against diphtheria in EU/EEA in the older age groups is suboptimal and certainly not sufficient, and might indicate a need for boosting immunity. It emphasises the potential risk of suboptimal protection against diphtheria in a time of high population mobility, outbreaks in certain parts of the world and the global shortage of diphtheria antitoxin.

In contrast with diphtheria, the seroprotection levels for tetanus were very reassuring leaving only 38 (3.7‰) sera without protective levels and in seven countries seroprotection was complete. The immunogenicity of the tetanus vaccine is superior to the diphtheria vaccine as reflected by higher vaccine-induced antibody responses to tetanus in numerous vaccine studies in all age groups[40,41]. The age and country effect on the seroprotection levels were similar to diphtheria. For sex the effect reached significance only at the protective cut-off in the whole study and within the countries there was also a trend towards higher levels of tetanus antibodies in females compared to males in 12/16 countries reaching significance in six countries, pointing to a sex-specific difference in immunity[9]. Overall, the protection against tetanus in EU/EEA needs no extra further action. Moreover, we found no correlation what so ever between the antibody levels of pertussis, diphtheria and tetanus with each other, indicating no interaction between these vaccines.

In conclusion, this cross-sectional retrospective seroprevalence study among middle-aged adults in 18 EU/EEA countries showed that there is circulation of *B. pertussis* despite highly implemented childhood vaccination programmes. Furthermore, it indicates a large underreporting of pertussis cases, also in the middle-aged population. Clearly the current monitoring system of pertussis based on case reporting is under-ascertaining disease, emphasising

that the current monitoring system of pertussis based on case reporting is not accurate enough and that sero-epidemiology is a valuable tool to monitor disease complementary to the current surveillance programmes. For diphtheria, the proportion of sera with non-protective IgG levels is of concern, leaving between 23 and 82% of the middle-aged population unprotected. Therefore, the protection against diphtheria in EU/EEA in these older age groups is not sufficient and deserves proper attention. In contrast to diphtheria, the seroprotection levels for tetanus were much higher, leaving only 3.7‰ of the sera with non-protective IgG levels. The seroprotection against tetanus is reassuring warranting no extra action but still requires ongoing monitoring to ensure this situation remains unchanged.

## Methods

**Study design and sample collection.** This study was organised by the National Institute of Public Health and the Environment (RIVM) and the University of Turku (UTU) under contract of ECDC/EUPert-LabNet (ECDC/2015/009). All national reference laboratories in EU/EEA countries were invited to participate. The study design was a random sample of retrospective collections of anonymised leftover serum samples from diagnostics or anonymised samples from nationwide serosurveillance studies with formal approval from a medical ethical committee (Medical Ethics Committee Noord-Holland, number: M015-022) during 2015–2018. No individual donor information was requested except age and date of sampling. As a strict inclusion criterium, serum samples should not have been collected from individuals suspected for respiratory infections, whereas samples from subjects with known respiratory infections do not reflect a "healthy" population. The successive age groups of 40–49 and 50–59 years were chosen for this study. As there are many boosters included in the national immunisation programmes, we decided to choose cohort of 40–59 years, which is less vulnerable for a bias caused by booster vaccinations and all significant antibody values in this study should reflect a true exposure to the pathogen. The recommended number of samples collected per country in each age group was 250 for a prevalence estimation of 20% with 95% confidence intervals (CI) in each age group independently of each other and regardless of country population. An equal distribution of sera from male and female donors was preferred.

**Procedures.** In this study, centralised testing for all collected samples was performed with the multiplex immunoassay (MIA). The use of the validated DTP-MIA showing 1:1 correlation with the original FDA-ELISA for pertussis and 1:1 correlation with the neutralisation assay for diphtheria[42–44] offered the advantage of measuring PT, Dt and TT antibodies simultaneously in one assay run[45,46]. In short, purified antigens were coupled covalently to distinct colour-coded activated carboxylated beads (Luminex, Austin, Texas, USA). Serum samples were measured in duplicate (1/200 and 1/4000 dilution), with in-house references calibrated against international standards, control sera and blanks included on each plate and MFI was converted to IU/mL by interpolation from a five-parameter logistic standard curve. As control for possible drift of the assay in time and different bead batches used, 5% of randomly selected samples per country were assessed in an extra duplicate measurement.

**Outcomes.** The first primary outcome was the proportion of samples from the middle-aged cohorts with a recent exposure to pertussis. As there has been discussions to include adult boosters into the European NIPs, we wanted to investigate what the pertussis burden in the selected age cohort is. We also know that subjects in this cohort have been immunised long time ago by primary wP vaccinations. In absence of an internationally accepted correlate of protection for pertussis, the cut-off value of IgG-PT ≥ 100 IU/mL was defined as indicative for a recent exposure and a level of ≥50 IU/mL for the prevalence of exposure 2 years prior the sample being taken, according to previous recommendations from several studies and from the European reference laboratories[11,14,47]. In addition, the proportion of sera with undetectable (lower limit of quantification (LLOQ) = 0.85 IU/mL) anti-PT IgG antibodies was used as an indication of lost antibody-based immunity to pertussis. The second primary study outcome was to estimate the level of vaccine-induced protection against diphtheria and tetanus based on the WHO cut-off level of 0.01 IU/mL for basic immunity and 0.1 IU/mL for protection for anti-diphtheria (IgG-Dt) and anti-tetanus (IgG-TT) antibodies[40,41]. The LLOQ for Dt and TT was 0.001 IU/mL.

**Statistical analysis.** The following information was collected: country, laboratory and location of sample collection, date of collection, sample size, age group, sex and type of collection. The final database included approximately 31,000 IgG results. Antibody levels below the LLOQ were replaced by LLOQ/2. The GMCs with 95% CI were estimated by linear regression modelling of the log-antibody levels, where age group, sex and country, up to their three-way interaction, were used as explanatory variables. Similarly, the seroprevalences with 95% CI were estimated by

binomial logistic regression modelling. The IgG-PT cut-off of 100 IU/mL is indicated to have 78% sensitivity and 98% specificity[14]. The corresponding seroprevalence estimates were adjusted accordingly using this formula: $p_{true} = (p_{est} + Sp - 1)/(Se + Sp - 1)$. Differences between GMCs and their corresponding $p$ values were obtained by least-squares means[48]. Similarly, this was done for the differences between seroprevalences. With regard to tetanus, for specific combinations of country, age group and sex only sero-negative outcomes were found. Therefore, seroprevalences were calculated with their corresponding 95% CI using the exact method[49] and $p$ values for differences in seroprevalences were calculated using Fisher's exact 2 × 2 test. All reported $p$ values are two-sided; $p$ values smaller than 0.05 are considered significant. Overall differences in seroprevalences between age groups, sex and countries were assessed using the likelihood ratio test, where the binomial logistic regression models from above were compared with a model without age group, sex or country, respectively. We arbitrarily subdivided the collection period into three groups (2015–2016, 2016–2017 and ≥2017) and for the geographical origin of the samples the countries were arbitrarily subdivided into three groups of one location, 2–7 locations or whole country to determine any possible statistical influence of these parameters. All statistical analyses were carried out in R (version 4.0.2)[50].

**Ethical approval.** The serosurveillance study in the Netherlands was approved by the Medical Ethics Committee Noord-Holland (METC number: M015-022). For the anonymised leftover samples from the other participating countries no ethical approval was required.

**Reporting summary.** Further information on research design is available in the Nature Research Reporting Summary linked to this article.

## Data availability

All relevant data are available online at https://github.com/kassteele/EU_Pertussis_seroprevalence.

## Code availability

IgG antibody levels against pertussis toxin, diphtheria toxoid and tetanus toxin in all serological samples were measured using the Bioplex LX200 and the software programme Bioplex Manager 6.2 (Bio-Rad Laboratories, Hercules, CA, USA). Further data collection has been done in Excel (Microsoft Office 365, version 16.0).

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

## Acknowledgements

We would like to thank from all participating countries the laboratories and personnel who contributed to the collection of the serum samples for this study, in particular Karin Baier (Austria), Amber Litzroth (Belgium), Sophie Guillot, Sabine Trombert-Paolantani, Véronique Jacomo (France), Paraskevi C. Fragkou, Theano Georgakopoulou (Greece), Erzsébet Rusvai (Hungary), Suzanne Cotter (Ireland), Jurijs Perevoscikovs (Latvia), Gaby Smits, Mark Jonker (the Netherlands), Cristina Oprea (Romania), Teodora Aktas, Teghesti Tecleab, Tiia Lepp (Sweden) and Ezra Linley (United Kingdom). The funder of the study had a role in the study design, data interpretation and writing of the manuscript, but no role in data collection or data analysis. The corresponding authors had full access to all data from the study, and final responsibility for the decision to submit for publication was by consensus of all co-authors. The study was undertaken as part of the framework contract "Coordination of activities for laboratory surveillance of whooping cough in Member States/EEA countries" (EUPert-LabNet) [ECDC/2015/009] coordinated by the National Institute for Public Health and the Environment (RIVM, the Netherlands) and the University of Turku (Finland) and funded by ECDC. Part of this research has been conducted using the Danish National Biobank resource, supported by the Novo Nordisk Foundation, grant number 2010-11-12 and 2009-07-28. The Dutch sample collection was commissioned and funded by the Ministry of Health, Welfare and Sport (VWS) of the Netherlands.

## Author contributions

Study concept and design: G.B., Q.H. and A.B. Data analysis and interpretation: G.B., P.v.G., J.v.d.K., Q.H. and A.B. Drafting of the manuscript: G.B., P.v.G., J.v.d.K., Q.H., A.B. Critical revision of the manuscript for intellectual content: U.W., I.D., T.D., J.T., S.T., I.P.F., K.M., A.G., T. Kolupajeva, D.F.V., P.P., O.P., L.W., T. Kastrin, L.M., H.C., C.K., S.B. and J.J. Statistical analysis: J.v.d.K. with input from G.B. and P.v.G. Obtained funding: Q.H. and S.B. Administrative, technical or material support: P.v.G. and Serosurveillance study team. Data acquisition: J.v.V., M. Bogaard and E.v.L. Supervision: G.B. and Q.H. All authors contributed to the interpretation of data and approved the final paper. The corresponding authors attest that all listed authors meet authorship criteria and that no others meeting the criteria have been omitted.

## Competing interests

The authors declare no competing financial interests.

## Additional information

## the Serosurveillance Study Team

Jeffrey van Vliet[1], Marjan Bogaard[1], Elske van Loghem[1], Fiona van der Klis[1], Joanna Jasinska[2], Denis Piérard[3], Charlotte Sværke Jørgensen[4], Sylvain Brisse[5], Vasiliki Rapti[6], Zsuzsanna Molnar[7], Deirdre Burke[8], Vilnele Lipnickiene[9], Jelena Galajeva[10], Audun Aase[11], Sofia Moura[12], Mihaela Leustean[13], Mia Brytting[14], Marta Vitek[15], Maria Avdicova[16], Gayatri Amirthalingam[17] & Jussi Mertsola[19]

