## [Peer Review File · Nature Communications]

Reviewer’s report on: Widespread circulation of pertussis and poor protection against diphtheria among middle-aged adults in 18 European countries

This paper presents the results of a large ($n \approx 10^4$) seroprevalence study conducted during 2015–2018 in 18 European countries, in the demographic of adults aged 40–59 years. The diseases examined are pertussis, diphtheria, and tetanus, all bacterial infections that are targeted by a combined vaccine (DTP), with historically large coverage throughout Europe. The results suggest that, among individuals aged 40–59, the circulation of pertussis is high (seroprevalence of 4–6%) and that a large fraction of individuals (range 20–80%) have antibody titers that may not be protective against diphtheria.

Although the study design and the methods are not particularly new, the strength of this work lies in its large sample size and geographical scope. Nevertheless, I have a number of substantive comments about the statistical analysis and the interpretation of the results. Please note that my comments focus on pertussis, about which I have published a number of papers recently [1, 2, 3, 4]. Because of my lesser knowledge of diphtheria epidemiology, I am less qualified to comment on those results, but the finding that a large fraction of adults have unprotective levels of antibodies is potentially highly significant. In the interest of transparency and openness, I am signing my review.

Dr. Matthieu Domenech de Cellès

Major comments

1. [Statistical analysis] As it stands, the statistical analysis is a bit weak and may not fully exploit the information present in this large data set. Indeed, for the most part, the analysis is based on a series of statistical tests, applied to data grouped by country, age (2 predefined age groups: 40–49 and 50–59), and gender. In the methods (p 7, lines 151–152), the authors indicate that they also used regression models, but it is not clear to me why the whole data have not been analyzed that way. It would be much more effective (and elegant) to apply a single logistic regression model to analyze the whole dataset, with a mixed-effects formulation to capture and to quantify the variability in seroprevalence by age, country, etc.
2. [Quality of model fit] Another weakness of the analysis is the absence of any measure of how well the statistical models fit the observed data. At least for pertussis, antibody titer data are indeed typically very noisy [5]. Related to my comment above, the use of logistic regression would allow to apply standard diagnostic methods. Given the large sample size, it could also be interesting to have a test set to diagnose the ability of the model to predict out-of-fit data.
3. [Low antibody titer interpreted as marker of susceptibility to pertussis] In the Methods (p 6, lines 136–137), the authors write: “In addition, the proportion of sera below the level of 5 IU/mL was used as indication of subjects potentially susceptible to pertussis.” The authors provide no reference to support this statement and seem to contradict themselves here, as they acknowledge somewhere else in the text (p 4, lines 95–96) that, as yet, no serological correlate has actually been found. I suggest removing all results and discussion related to this arbitrary threshold.
4. [High antibody titers interpreted as marker of recent pertussis infection] More to the point, the authors interpret antibody titers above 100 IU/mL as a marker of recent pertussis infection. Although many other studies have also used that threshold, the authors should be aware that such an interpretation may also be problematic. Specifically, a rise in antibody titer (not associated with clinical symptoms) may simply reflect an anamnestic response, without the inception of transmissible infection [6]—that is, it may not represent a “real” infection, but simply an exposure to pertussis. These concerns are not new; already in 1992 Noel Preston wrote in the *Lancet* [7]: “The diagnosis of pertussis is in serious danger of derailment at a time when its reliability is more important than ever. Despite increasing use of serology, the gold standard for laboratory diagnosis of pertussis is bacterial culture from nasopharyngeal swabs.” Sadly, almost 30 years after that statement, the situation has not improved and many investigators still blindly rely only on serological studies to evaluate the burden of pertussis. In addition, there is now a large body of research showing—contrary to widespread beliefs in the field—that pertussis vaccines confer waning, but long-term immunity, that pertussis circulation is low overall (and in particular in adults), and that pertussis resurgence may not be the consequence of bad vaccines [1, 2, 3, 4, 8, 9, 10, 11, 12, 13, 14].

That the authors do not cite this evidence, or do not make any effort to contextualize their results, is regrettable. More generally, ignoring counter-evidence is not a sound scientific practice.

5. [Temporal variations in pertussis seroprevalence] In the discussion (p10, lines 236–239), the authors write that they found no statistical evidence for temporal variations in pertussis seroprevalence. Considering the marked periodicity of pertussis (with peaks every 4–5 years in the vaccine era [15]), and as the study period roughly encompasses a whole epidemic cycle, this finding is surprising. It would be interesting to estimate in more detail the temporal trends (e.g., using a flexible function of time, incorporated into the logistic regression model, cf. my comment above) in each country, and to compare them with reported incidence data. I suspect the discrepancy will be large, and in any case worth commenting upon. Admittedly, pertussis incidence data are under-reported and do not provide the full picture, but as I wrote above there is no reason to believe that serological data provide some kind of “truth” (indeed, there is much evidence against it).

Minor comments

- The authors should better explain the choice of the study population. Why focus on the particular age group of 40–59?
- Introduction, p 4 line 89: the term “pressure of infection” is a bit vague and may be confused with the term “force of infection”, used in epidemiological modeling. Maybe write “burden of infection” instead?
- Introduction, p 4 line 9: “Furthermore, seroprevalence studies offer an opportunity to study waning immunity in the population.” This may be true for diphtheria and tetanus, but not for pertussis, inasmuch as the current absence of serological correlate of protection does not allow to correlate an antibody titer to a level of protection.
- Methods, p 6 lines 116–117: “Serum samples should not have been collected from individuals suspected for respiratory-infections”. Why not? As noted in my comment above, the presence of respiratory symptoms—in addition to a rise in antibody titers—would provide some reassurance that the detected infection is real (that is, transmissible) [6, 7].
- Discussion, p 12, lines 278–280: “To get a better estimate of the circulation of *B. pertussis* in the population, (regular) sero-epidemiology is a valuable tool complementary to surveillance programs based on case reporting.” Please see my comments above. On the contrary, the two approaches have led to widely different estimates of pertussis burden, so that it is hard to understand how they could be considered complementary.

References

- [1] Domenech de Cellès M, Riolo MA, Magpantay FMG, Rohani P, King AA. Epidemiological evidence for herd immunity induced by acellular pertussis vaccines. *Proc Natl Acad Sci U S A*. 2014 Feb;111(7):E716–7.
- [2] Domenech de Cellès M, Magpantay FMG, King AA, Rohani P. The pertussis enigma: reconciling epidemiology, immunology and evolution. *Proc Biol Sci*. 2016 Jan;283(1822).
- [3] Domenech de Cellès M, Magpantay FMG, King AA, Rohani P. The impact of past vaccination coverage and immunity on pertussis resurgence. *Sci Transl Med*. 2018 03;10(434).
- [4] Domenech de Cellès M, Rohani P, King AA. Duration of Immunity and Effectiveness of Diphtheria-Tetanus-Acellular Pertussis Vaccines in Children. *JAMA Pediatr*. 2019 06;173(6):588–594.
- [5] Versteegh FGA, Mertens PLJM, de Melker HE, Roord JJ, Schellekens JFP, Teunis PFM. Age-specific long-term course of IgG antibodies to pertussis toxin after symptomatic infection with *Bordetella pertussis*. *Epidemiol Infect*. 2005 Aug;133(4):737–48.
- [6] Fine PE. Adult pertussis: a salesman’s dream—and an epidemiologist’s nightmare. *Biologicals*. 1997 Jun;25(2):195–8.
- [7] Pertussis: adults, infants, and herds. *Lancet*. 1992 Feb;339(8792):526–7.

- [8] Rohani P, Earn DJ, Grenfell BT. Impact of immunisation on pertussis transmission in England and Wales. *Lancet*. 2000 Jan;355(9200):285–6.
- [9] Rohani P, Zhong X, King AA. Contact network structure explains the changing epidemiology of pertussis. *Science*. 2010 Nov;330(6006):982–5.
- [10] Blackwood JC, Cummings DAT, Broutin H, Iamsirithaworn S, Rohani P. Deciphering the impacts of vaccination and immunity on pertussis epidemiology in Thailand. *Proc Natl Acad Sci U S A*. 2013 Jun;110(23):9595–600.
- [11] Riolo MA, King AA, Rohani P. Can vaccine legacy explain the British pertussis resurgence? *Vaccine*. 2013 Dec;31(49):5903–8.
- [12] Préziosi MP, Halloran ME. Effects of pertussis vaccination on transmission: vaccine efficacy for infectiousness. *Vaccine*. 2003 May;21(17-18):1853–61.
- [13] Taranger J, Trollfors B, Bergfors E, Knutsson N, Sundh V, Lagergård T, et al. Mass vaccination of children with pertussis toxoid—decreased incidence in both vaccinated and nonvaccinated persons. *Clin Infect Dis*. 2001 Oct;33(7):1004–10.
- [14] Trollfors B, Taranger J, Lagergård T, Sundh V, Bryla DA, Schneerson R, et al. Immunization of children with pertussis toxoid decreases spread of pertussis within the family. *Pediatr Infect Dis J*. 1998 Mar;17(3):196–9.
- [15] Broutin H, Viboud C, Grenfell BT, Miller MA, Rohani P. Impact of vaccination and birth rate on the epidemiology of pertussis: a comparative study in 64 countries. *Proc Biol Sci*. 2010 Nov;277(1698):3239–45.

Reviewer #2 (Remarks to the Author):

The paper by Berbers and coworkers reports seroprevalence of antibodies anti-pertussis, diphtheria and tetanus in the age groups 40-49 and 50-59 in 18 EU/EEA countries. The results obtained highlight a widespread and homogenous circulation of *B. pertussis* and poor protection against diphtheria. These notions should be carefully taken into consideration, especially by public health authorities.

The manuscript is well and clearly written. The conclusions are sound.

Minor:

Lines 100-101: the pertussis vaccine also has a longstanding story and high coverage (they are given together), hence it should be mentioned that those vaccines probably confer a better protection than the pertussis one.

Reviewer #3 (Remarks to the Author):

This is a very nice international seroprevalence study on Diphtheria, pertussis and tetanus. Sometimes details are missing for a better understanding of the methodology.

A few general remarks:

How would you in general differentiate those with recent vaccination from those with infection in the last 2 years?

Did you correlate the titers between the different antibodies? for example did the unprotected for diphtheria also have lower tetanus levels?

details:

Line 114: retrospective collected leftover samples from what kind of studies other than seroprevalence studies?

Line 117: Why did you choose these age categories?

Line 135: What is meant by recent?

Line 135-136: Recent vaccination, how to differentiate from infection in 2 years prior to sample taken?

Line 144: What do you mean with type of collection?

Line 164: How did you handle the datasets without gender?

Line 145 mentions 31000 results, line 166 mentions 10302 results? What happened in the selection procedure and why do you call the 31000 results the final database?

Line 170: Intermediate levels high in Norway, vaccination programs?

Line 238: subdivisions of countries and time periods is not discussed in the methods or results section; could you please elaborate in the respective sections to introduce these discussion items

Line 272: this refers to the same references as mentioned at the start of your introduction. Maybe be more precise in the introduction about the 500/100000 and not >100

Line 294 and following: Since the infection is toxin mediated, protection will most probably be through the antibodies of course. But could it be that another immunological mechanism could play a role, eg T cell responses? Susceptibility is very high in some groups yet no major outbreaks occur. An interesting other source could be a link to the European diphtheria surveillance network. They published last year a nice manuscript on Diphtheria. Naresh Chand Sharma, Androulla Efstratiou, Igor Mokrousov, Ankur Mutreja, Bhabatosh Das & Thandavarayan Ramamurthy. Nature Reviews Disease Primers volume 5, Article number: 81 (2019). These authors refer to the gender effect in protection from diphtheria as well as on the low rate of serological immunity among adults depending on the time delay in vaccination during their childhood, that is, a delay up to 3 years.

Reviewer #1 (Remarks to the Author):

Reviewer's report on: Widespread circulation of pertussis and poor protection against diphtheria among middle-aged adults in 18 European countries

This paper presents the results of a large ($n \sim 10^4$) seroprevalence study conducted during 2015–2018 in 18 European countries, in the demographic of adults aged 40–59 years. The diseases examined are pertussis, diphtheria, and tetanus, all bacterial infections that are targeted by a combined vaccine (DTP), with historically large coverage throughout Europe. The results suggest that, among individuals aged 40–59, the circulation of pertussis is high (seroprevalence of 4–6%) and that a large fraction of individuals (range 20–80%) have antibody titers that may not be protective against diphtheria. Although the study design and the methods are not particularly new, the strength of this work lies in its large sample size and geographical scope. Nevertheless, I have a number of substantive comments about the statistical analysis and the interpretation of the results. Please note that my comments focus on pertussis, about which I have published a number of papers recently [1, 2, 3, 4]. Because of my lesser knowledge of diphtheria epidemiology, I am less qualified to comment on those results, but the finding that a large fraction of adults have unprotective levels of antibodies is potentially highly significant. In the interest of transparency and openness, I am signing my review.

Dr. Matthieu Domenech de Cellès

Major comments

1. [Statistical analysis] As it stands, the statistical analysis is a bit weak and may not fully exploit the information present in this large data set. Indeed, for the most part, the analysis is based on a series of statistical tests, applied to data grouped by country, age (2 predefined age groups: 40–49 and 50–59), and gender. In the methods (p 7, lines 151–152), the authors indicate that they also used regression models, but it is not clear to me why the whole data have not been analyzed that way. It would be much more effective (and elegant) to apply a single logistic regression model to analyze the whole dataset, with a mixed-effects formulation to capture and to quantify the variability in seroprevalence by age, country, etc.

We would like to thank the reviewer for his suggestions. Indeed, most part of the analysis was a series of statistical tests: t-tests for log-GMC ratios and exact tests (Fisher's test) for the prevalence data given as counts. In our opinion these tests were justified, because they answered the questions being asked, such as: is there, within one country, given a certain age group, a difference between males and females? Besides, the exact method did not give any issues with complete separability, e.g. having only sero-negative outcomes in one group. The reviewer correctly suggests that the same questions could be answered by regression modelling. Using a linear model for the log-GMCs and a binomial regression model (aka logistic regression model) for the counts, and including full three-way interaction between country, age group and gender, we obtain almost the same results. Almost, because for the linear model, the residual degrees of freedom becomes much larger as all observations are pooled together in one analysis and the residual variance is assumed to be constant over all observations. For the binomial model, the statistical tests are not exact anymore. Instead, we obtain a Wald approximation. Given the large sample sizes, this should not be an issue. For tetanus

we had to make an exception. For specific combinations of country, age group and gender, only sero-negative outcomes were found, which resulted in infinite confidence intervals. Here we still applied the exact method. We have enclosed the results of the regression model for tetanus hereunder.

Throughout the manuscript we have replaced the results of the series of statistical tests by the results of the above regression models as described in M&M on lines 156-161. As stated above, the results are almost identical. We could however not fulfill the reviewer's suggestion to include country as a random effect for three reasons. First, including country as a random intercept would lead to a too simplistic model. The effects of age group and gender differ between countries. This directly brings us to the second point, including country as a random slope for age group and gender. This appears to be impossible because these models fail to converge. The third reason not to use a mixed model is that it does not answer the question above. A mixed model induces shrinkage towards a "mean country". Such estimates are not of interest here.

2. [Quality of model fit] Another weakness of the analysis is the absence of any measure of how well the statistical models fit the observed data. At least for pertussis, antibody titer data are indeed typically very noisy [5]. Related to my comment above, the use of logistic regression would allow to apply standard diagnostic methods. Given the large sample size, it could also be interesting to have a test set to diagnose the ability of the model to predict out-of-fit data.

As the reviewer suggests, the regression modelling approach allows us to apply standard diagnostic methods. Besides, we fully agree with the reviewer's remark that antibody concentration data are very

noisy. Conditional on country, age group and gender, there is still very much between-person variation that we cannot account for. It then becomes challenging to come up with a 'good' model. However, if our goal is to estimate GCM's and prevalence's, and answer questions about differences between groups, such as above, both the linear model as well as the binomial regression are perfectly able to do this. In other words, even if the model fit is poor, that does not mean the model is useless. Our goal is to estimate group means and prevalence's. If we have a fully saturated model (i.e. the three-way interaction between country, age group and gender), there is nothing to improve on. We therefore think it is of no additional value to provide a goodness-of-fit statistic, because it is not relevant in this context.

3. [Low antibody titer interpreted as marker of susceptibility to pertussis] In the Methods (p 6, lines 136–137), the authors write: “In addition, the proportion of sera below the level of 5 IU/mL was used as indication of subjects potentially susceptible to pertussis.” The authors provide no reference to support this statement and seem to contradict themselves here, as they acknowledge somewhere else in the text (p 4, lines 95–96) that, as yet, no serological correlate has actually been found. I suggest removing all results and discussion related to this arbitrary threshold.

*We agree that the 5 IU/mL cut-off is misleading for the statement “potentially susceptible subjects”. This, as you pointed out, is against the current situation where no serological correlate has been defined. Therefore, we have replaced this cut-off with undetectable (LLOQ = 0.85 IU/mL) anti-PT IgG antibodies, which reflects the number of subjects with no detectable antibodies against PT and therefore assumable they do not have protection when measured by the circulating anti-PT antibodies. This does not mean that they do not have e.g. cell-mediated immunity or antibodies against other antigens of *B. pertussis*. Because this study is only focused on the anti-PT antibodies as they are specific to *B. pertussis* and we do not have any other immunological information from the subjects, except the fact that they have not had any respiratory symptoms. We have corrected this information on lines 148-152 and 187-188 and amended the text throughout the manuscript and in Table 2 and Figure 1.*

4. [High antibody titers interpreted as marker of recent pertussis infection] More to the point, the authors interpret antibody titers above 100 IU/mL as a marker of recent pertussis infection. Although many other studies have also used that threshold, the authors should be aware that such an interpretation may also be problematic. Specifically, a rise in antibody titer (not associated with clinical symptoms) may simply reflect an anamnestic response, without the inception of transmissible infection [6]—that is, it may not represent a “real” infection, but simply an exposure to pertussis. These concerns are not new; already in 1992 Noel Preston wrote in the Lancet [7]: “The diagnosis of pertussis is in serious danger of derailment at a time when its reliability is more important than ever. Despite increasing use of serology, the gold standard for laboratory diagnosis of pertussis is bacterial culture from nasopharyngeal swabs.” Sadly, almost 30 years after that statement, the situation has not improved and many investigators still blindly rely only on serological studies to evaluate the burden of pertussis. In addition, there is now a large body of research showing—contrary to widespread beliefs in the field—that pertussis vaccines confer waning, but long-term immunity, that pertussis circulation is low overall (and in particular in adults), and that pertussis resurgence may not be the consequence of bad vaccines [1, 2, 3, 4, 8, 9, 10, 11, 12, 13, 14]. That the authors do not cite this evidence, or do not make any effort to contextualize their results, is regrettable. More generally, ignoring counter-evidence is not a sound scientific practice.

We agree that this issue might be problematic. However, as stated also by the reviewer this cut-off has been widely used and a recommendation to use this cutoff was published “Guiso et al. What to do and what not to do in serological diagnosis of pertussis: recommendations from EU reference laboratories” (ref #. 14 in this manuscript) in which a table with recommended cut offs from Europe, the US and Australia can be found. We have added this including the reference on line 147. Since booster doses in different age groups are used in different countries, pertussis vaccination may interfere with the interpretation of serological results, the vaccination status of study subjects is also taken into account, subjects should not have received a booster dose within a year. This interpretation is also based on several studies in which decay of anti-PT antibodies is found to be rather fast as shown in the publications by Hans Hallander (Sweden), Dalby (Denmark) and Teunis (the Netherlands). We have added a sentence with these references to the manuscript to highlight this issue. Lines 301-303.

Regarding the recent infection versus exposure. In this study, we aimed to detect the circulation of B. pertussis among healthy population and therefore both options actually means the same thing. On several places in the manuscript we have changed infection in exposure. If there is no exposure to the pathogen, the options to have highly elevated antibodies can come either from a recent vaccination or from cross-reactivity of antibodies against another protein. However, the latter option is very unlikely. In addition, we do not have any sera from subjects with respiratory symptoms, which could lead to a higher proportion of exposed individuals. We also agree that the culture is gold standard for diagnosis of pertussis. However, the sensitivity of this method is relatively low, especially for adults. If we only rely on culture, the number of reported cases would be extremely low.

Regarding the counter-evidence, we agree that we have not considered all aspects. We have thus added a paragraph in the discussion section to highlight findings from mathematical models as suggested by the reviewer. We also added several references as suggested. In addition, we have added that other immunological responses also affects to the entity of immunological protection acquired from the vaccination, not only anti-PT antibodies. Furthermore, antibodies against other B. pertussis antigens may also confer some protection. It is known that using only antibody responses as an indicator of immune responses can skew the whole picture. Therefore we have added sentences that underline this issue and that further investigations are needed to conclude the real long-term protection. Lines 283-287 and lines 317-326.

5. [Temporal variations in pertussis seroprevalence] In the discussion (p10, lines 236–239), the authors write that they found no statistical evidence for temporal variations in pertussis seroprevalence. Considering the marked periodicity of pertussis (with peaks every 4–5 years in the vaccine era [15]), and as the study period roughly encompasses a whole epidemic cycle, this finding is surprising. It would be interesting to estimate in more detail the temporal trends (e.g., using a flexible function of time, incorporated into the logistic regression model, cf. my comment above) in each country, and to compare them with reported incidence data. I suspect the discrepancy will be large, and in any case worth commenting upon. Admittedly, pertussis incidence data are under-reported and do not provide the full picture, but as I wrote above there is no reason to believe that serological data provide some kind of “truth” (indeed, there is much evidence against it).

We did not find any evidence for a period effect. The main reason is the lack of proper information about the date of sampling. We only have rough time periods encompassing several years mostly. As a consequence, possible periodic effects average out. The reviewer suggests to include time as a spline

curve in the regression model. We have tried this, but the model failed to converge. This leaves us with no other option than to use the three time periods we have used before.

It is true that pertussis has been considered as a cyclic disease. However, there has been inclusion of e.g. adolescent boosters, which have clearly affected the circulation of the disease. In the article by de Cellès, “The impact of past vaccination coverage and immunity on pertussis resurgence”, the impact of child and adolescent boosters have been highlighted as they should decrease the transmission of the disease more than adult boosters. As there is less circulation, there will be fewer infections. Furthermore, nationwide epidemics in many European countries before this study have definitively had an impact on the results of this study. Regarding the incidence data during the study period, we agree that adding the incidence data for comparison would be interesting and we have added the notification rates/100,000 (subjects ≥15y, ECDC surveillance atlas) during 2015-2018 by country into supplemental table 2. See also line 305.

Minor comment

- The authors should better explain the choice of the study population. Why focus on the particular age group of 40–59?

As there are many boosters included in the national immunization programs (NIP), we decided to choose cohort of 40-59 years, which is less vulnerable for a bias caused by booster vaccinations. This means that in general, all significant antibody values in this study should reflect a true exposure to the pathogen. As there has also been discussions to include adult boosters into the NIPs, we wanted to investigate what the pertussis burden in this age cohort is. We also know that subjects in this cohort have been immunized by primary wP vaccination. We have added this information into the text on lines 120-126, to make it more clear for the reader, why this cohort was selected. Furthermore we added two sentences to clarify the primary outcome on lines 141-144.

- Introduction, p 4 line 89: the term “pressure of infection” is a bit vague and may be confused with the term “force of infection”, used in epidemiological modeling. Maybe write “burden of infection” instead?

We agree with the reviewer and changed this accordingly.

- Introduction, p 4 line 9: “Furthermore, seroprevalence studies offer an opportunity to study waning immunity in the population.” This may be true for diphtheria and tetanus, but not for pertussis, inasmuch as the current absence of serological correlate of protection does not allow to correlate an antibody titer to a level of protection.

As stated in the response to major comment 3, we changed the 5 IU/mL into the LLOQ value (0.85 IU/mL); in this way we will determine those individuals with no detectable antibodies against PT left in their blood. However, this does not exactly answer to your comment, so we have modified the sentence a bit to show that this estimation is only based on the anti-PT antibody results. As we do not have any other immunological information available for the subjects, we cannot conclude that there is no protection left. However, we do not totally agree that you cannot use antibody concentrations as a marker of waning immunity as it at least shows one side of the story. Line 91-95.

- Methods, p 6 lines 116–117: “Serum samples should not have been collected from individuals suspected for respiratory-infections”. Why not? As noted in my comment above, the presence of respiratory symptoms—in addition to a rise in antibody titers—would provide some reassurance that the detected infection is real (that is, transmissible) [6, 7].

We understand this comment, but to include these subjects would be against the inclusion criteria from the study, which state amongst others that sera should be selected from individuals with no suspected respiratory infections. We cannot change the inclusion criteria, but we will highlight this issue better in the text as it seems to be poorly presented and may raise questions from the readers. Lines 121-122.

- Discussion, p 12, lines 278–280: “To get a better estimate of the circulation of B. pertussis in the population, (regular) sero-epidemiology is a valuable tool complementary to surveillance programs based on case reporting.” Please see my comments above. On the contrary, the two approaches have led to widely different estimates of pertussis burden, so that it is hard to understand how they could be considered complementary.

We agree that this sentence may be misleading. We have modified the sentence and offered explanations in the extra paragraph dedicated originally to the major comment 4.

Reviewer #2 (Remarks to the Author):

The paper by Berbers and coworkers reports seroprevalence of antibodies anti-pertussis, diphtheria and tetanus in the age groups 40-49 and 50-59 in 18 EU/EEA countries. The results obtained highlight a widespread and homogenous circulation of B. pertussis and poor protection against diphtheria. These notions should be carefully taken into consideration, especially by public health authorities.

The manuscript is well and clearly written. The conclusions are sound.

Minor:

Lines 100-101: the pertussis vaccine also has a longstanding story and high coverage (they are given together), hence it should be mentioned that those vaccines probably confer a better protection than the pertussis one.

We thank the reviewer for his kind words. The reviewer is right that the D and T vaccines are mostly administered together with aP vaccines and earlier with the wP vaccines except for adult and adolescents in this case. The lack of wP vaccine boosters in earlier times could also have contributed to a less good protection against pertussis but the suggestion of the reviewer that these vaccines confer a better protection than the pertussis vaccines has been added. Line 104.

Reviewer #3 (Remarks to the Author):

This is a very nice international seroprevalence study on Diphtheria, pertussis and tetanus.

Sometimes details are missing for a better understanding of the methodology.

A few general remarks:

How would you in general differentiate those with recent vaccination from those with infection in the last 2 years?

It is not possible to differentiate between a recent pertussis vaccination and a pertussis infection in the last two years but a recent pertussis vaccination is not very likely in these age cohorts (40-59 years) because such adult boosters are not included in the national immunisation programmes except for France and Austria and the coverage of these adult boosters is very low. Moreover one of the exclusion criteria of the study is a recent pertussis vaccination. We have added a sentence about this issue on lines 287-290.

Did you correlate the titers between the different antibodies? for example did the unprotected for diphtheria also have lower tetanus levels?

We did not correlate the concentrations of the different vaccine antibodies with each other because we knew from earlier studies that these levels do not correlate. Here we show the 3 correlation graphs for 7 countries which clearly illustrate that there is no real correlation between these antibody levels. For some participants a unprotected diphtheria antibody level correlates with a very low tetanus level but for the majority this is not the case. We have added a sentence at the end of the discussion emphasizing this on line 354-355.

Details:

Line 114: retrospective collected leftover samples from what kind of studies other than seroprevalence studies?

In most countries the retrospective collected samples were diagnostic samples as mentioned in Table 1. We have added "diagnostics" on line 118.

Line 117: Why did you choose these age categories?

As there are many boosters included in the national immunization programs (NIP), we decided to choose cohort of 40-59 years, which is less vulnerable for a bias caused by booster vaccinations. This

means that in general, all significant antibody values in this study should reflect a true exposure to the pathogen. As there has also been discussions to include adult boosters into the NIPs, we wanted to investigate what the pertussis burden in this age cohort is. We also know that subjects in this cohort have been immunized by primary wP vaccination. We have added this information into the text on lines 120-126, to make it more clear for the reader, why this cohort was selected. Furthermore we added two sentences to clarify the primary outcome on lines 141-144.

Line 135: What is meant by recent?

For this arbitrarily chosen cut-off of 100 IU/mL recent means mostly an pertussis exposure in the last year but sometimes it is also used for 0.5 year. Therefore we like to use the term recent.

Line 135-136: Recent vaccination, how to differentiate from infection in 2 years prior to sample taken?

As explained here above with the general remarks it is not possible to differentiate between recent vaccination and infection in the last 2 years (please see above).

Line 144: What do you mean with type of collection?

Type of collection means either left-over samples from diagnostic procedures or samples from a seroprevalence study as explained in Table 1 and in Materials and methods on lines 117-119.

Line 164: How did you handle the datasets without gender?

We included the data sets from the two countries without gender data in the analyses but of course there was no outcome of gender statistics for these 2 countries.

Line 145 mentions 31000 results, line 166 mentions 10302 results? What happened in the selection procedure and why do you call the 31000 results the final database?

On line 183 we mention the participation of 10,302 subjects. As we received from each subject one serum samples in which we measured 3 different vaccine IgG antibody concentrations, we obtained 30,906 results. We mentioned on line 156 approximately 31,000 results.

Line 170: Intermediate levels high in Norway, vaccination programs?

The possible reasons for these high intermediate levels in Norway are not known. It might be that the vaccination program has had influence on these levels but that seems not so likely due to the older age of our participants. Another explanation might be the higher circulation of Bordetella in Norway

considering the high incidence numbers in Norway on the ECDC website for many years now. This might mean that people from these age groups have a higher chance of exposure to the pathogen.

Line 238: subdivisions of countries and time periods is not discussed in the methods or results section; could you please elaborate in the respective sections to introduce these discussion items

We have added a sentence for this arbitrarily chosen subdivision in the Materials and Methods section on lines 167-170.

Line 272: this refers to the same references as mentioned at the start of your introduction. Maybe be more precise in the introduction about the 500/100000 and not >100

We thank the reviewer for this observation; we have changed this in notification rate up to 50/100,000 according to the ECDC atlas for Pertussis on line 83 and 305.

Line 294 and following: Since the infection is toxin mediated, protection will most probably be through the antibodies of course. But could it be that another immunological mechanism could play a role, e.g. T cell responses? Susceptibility is very high in some groups yet no major outbreaks occur. An interesting other source could be a link to the European diphtheria surveillance network. They published last year a nice manuscript on Diphtheria. Naresh Chand Sharma, Androulla Efstratiou, Igor Mokrousov, Ankur Mutreja, Bhabatosh Das & Thandavarayan Ramamurthy. Nature Reviews Disease Primers volume 5, Article number: 81 (2019). These authors refer to the gender effect in protection from diphtheria as well as on the low rate of serological immunity among adults depending on the time delay in vaccination during their childhood, that is, a delay up to 3 years.

Of course cellular immunity plays also a very important role in immunity and therefore protection against pertussis by IgG-PT antibody levels is just a part of the whole story. But for obvious reasons we were not able to investigate these kind of immunological parameters in this study.

We thank the reviewer for the reference of the nice publication of the diphtheria network and we have added this observation with the reference on lines 339-340.

Reviewer’s report on: “Widespread circulation of pertussis and poor protection against diphtheria among middle-aged adults in 18 European countries”

I thank the authors for their thorough responses. The revised manuscript is now clearer and better emphasizes the caveats that concern serological studies for pertussis. Nevertheless, I still have a number of substantive comments that need to be addressed.

Major comments

1. The authors interpret high PT antibody titers as evidence of recent infection (or exposure), but it would be useful to clarify what they mean by that exactly and how what is estimated relates to standard epidemiological quantities (like prevalence or incidence of infection). Tools have been developed (by Peter Teunis and colleagues) and implemented in R (<https://cran.r-project.org/web/packages/seroincidence/index.html>) to estimate sero-incidence rates from cross-sectional data like those in this study. A second step would be to translate those sero-incidence estimates into estimates of sero-prevalence (that is, of the proportion of individuals infectious at a given time point). It is not easy to connect these two quantities in general, but even a rough approximation like $p = \Lambda D$ (where Λ is the sero-incidence rate estimate and D the infectious period of pertussis, approximately 3 weeks) would be useful. To illustrate, if we assume that the proportion with anti-PT IgG ≥ 100 IU/mL estimated in this study (denoted by f) represents infection in the past 6 months [1], then approximately $\Lambda = 2f$ per year (using the tools of the seroincidence R package will provide a much better estimate). Hence, $p \approx \Lambda D = 2 \times \frac{3}{52} f \approx 0.12f$, and this simple calculation suggests that the prevalence of infection is an order of magnitude smaller than the fraction estimated in this study.
2. In their response, the authors cite a paper about recommended cut-offs for serological diagnosis of pertussis [2]. In that paper (Table 2), the IgG anti-PT cutoff of 100 IU/mL is indicated to have $S_e = 78\%$ sensitivity and $S_p = 98\%$ specificity. Using these values to interpret (and adjust) the results of this study would be informative. Specifically, one can use the standard formula relating the true prevalence of infection p_{true} to the estimated prevalence p_{est} : $p_{\text{est}} = S_e p_{\text{true}} + (1 - S_p)(1 - p_{\text{true}})$. Application of this formula for $p_{\text{true}} = 0$ (i.e., no circulation of pertussis) and using the values of S_e and S_p indicated above, one finds that $p_{\text{est}} = 1 - S_p = 0.02$. Under the generous assumption that $f = p_{\text{est}}$ (in reality the estimated prevalence of infection will be much smaller than that, cf. my first comment), this means that estimates below 2% in this study (like in Finland, cf. Table 2) are in fact consistent with no circulation of pertussis at all (!). For values above 2%, one can inverse the formula to adjust the prevalence estimate: $p_{\text{true}} = \frac{p_{\text{est}} + S_p - 1}{S_e + S_p - 1}$. Applying this formula for p_{est} in the range 4–6.4% (as indicated in the abstract) suggests that the true prevalence is over-estimated by up to 50% in this study. Again, this is under the generous assumption that $f = p_{\text{est}}$; the simple calculation above suggests that p_{est} will actually be much smaller, so that in many countries (not just Finland) it will likely fall into the interval 0–2% associated with no circulation of pertussis (because of the imperfect specificity of serological testing).

Hence, once imperfect sensitivity and specificity are taken into account, the results of this study appear to indicate very little circulation of pertussis and the authors’ claim that pertussis circulation is widespread cannot be accepted until further analyses (like those proposed above) are conducted.

Minor comments

- The authors write that “[...] antibody concentrations as a marker of waning immunity [...] at least shows one side of the story.” I disagree. As already commented, no serological correlate of protection against pertussis has yet been found. Elsewhere in their replies, the authors comment on the fact that serum antibody levels quickly drop to undetectable levels after vaccination. This is true (see, for example, Ref. [3]), but then one wonders how serum antibodies can be considered as a marker of immunity, given that protection persists even after they have disappeared.

References

- [1] Noel G, Badmasti F, Nikbin VS, Zahraei SM, Madec Y, Tavel D, et al. Transversal sero-epidemiological study of *Bordetella pertussis* in Tehran, Iran. PLoS One. 2020;15(9):e0238398.

- [2] Guiso N, Berbers G, Fry NK, He Q, Riffelmann M, Wirsing von König CH, et al. What to do and what not to do in serological diagnosis of pertussis: recommendations from EU reference laboratories. *Eur J Clin Microbiol Infect Dis.* 2011 Mar;30(3):307–12.
- [3] Gustafsson L, Hallander HO, Olin P, Reizenstein E, Storsaeter J. A controlled trial of a two-component acellular, a five-component acellular, and a whole-cell pertussis vaccine. *N Engl J Med.* 1996 Feb;334(6):349–55.

Reviewer 1 remarks are in red

Reviewer #3 (Remarks to the Author):

This is a very nice international seroprevalence study on Diphtheria, pertussis and tetanus.

Sometimes details are missing for a better understanding of the methodology.

A few general remarks:

How would you in general differentiate those with recent vaccination from those with infection in the last 2 years?

It is not possible to differentiate between a recent pertussis vaccination and a pertussis infection in the last two years but a recent pertussis vaccination is not very likely in these age cohorts (40-59 years) because such adult boosters are not included in the national immunisation programmes except for France and Austria and the coverage of these adult boosters is very low. Moreover one of the exclusion criteria of the study is a recent pertussis vaccination. We have added a sentence about this issue on lines 287-290.

OK. It seems indeed unlikely that high antibody titers could correspond to a recent vaccination in that age group.

Did you correlate the titers between the different antibodies? for example did the unprotected for diphtheria also have lower tetanus levels?

We did not correlate the concentrations of the different vaccine antibodies with each other because we knew from earlier studies that these levels do not correlate. Here we show the 3 correlation graphs for 7 countries which clearly illustrate that there is no real correlation between these antibody levels. For some participants a unprotected diphtheria antibody level correlates with a very low tetanus level but for the majority this is not the case. We have added a sentence at the end of the discussion emphasizing this on line 354-355.

OK. The graphs indeed show no correlation, but it may nevertheless be interesting to add those in the supplement.

Details:

Line 114: retrospective collected leftover samples from what kind of studies other than seroprevalence studies?

In most countries the retrospective collected samples were diagnostic samples as mentioned in Table 1. We have added "diagnostics" on line 118.

OK.

Line 117: Why did you choose these age categories?

As there are many boosters included in the national immunization programs (NIP), we decided to choose cohort of 40-59 years, which is less vulnerable for a bias caused by booster vaccinations. This means that in general, all significant antibody values in this study should reflect a true exposure to the pathogen. As there has also been discussions to include adult boosters into the NIPs, we wanted to investigate what the pertussis burden in this age cohort is. We also know that subjects in this cohort have been immunized by primary wP vaccination. We have added this information into the text on lines 120-126, to make it more clear for the reader, why this cohort was selected. Furthermore we added two sentences to clarify the primary outcome on lines 141-144.

I also made this comment and their answer is reasonable.

Line 135: What is meant by recent?

For this arbitrarily chosen cut-off of 100 IU/mL recent means mostly an pertussis exposure in the last year but sometimes it is also used for 0.5 year. Therefore we like to use the term recent.

This is a critical comment, which is not properly addressed in my opinion. In my second review, I proposed further analyses to connect the proportions estimated in this study to more meaningful epidemiological quantities like prevalence or incidence rate. Doing so would considerably strengthen the paper and facilitate the interpretation of the results.

Line 135-136: Recent vaccination, how to differentiate from infection in 2 years prior to sample taken?

As explained here above with the general remarks it is not possible to differentiate between recent vaccination and infection in the last 2 years (please see above).

same question as above; the author's answer is OK.

Line 144: What do you mean with type of collection?

Type of collection means either left-over samples from diagnostic procedures or samples from a seroprevalence study as explained in Table 1 and in Materials and methods on lines 117-119.

OK.

Line 164: How did you handle the datasets without gender?

We included the data sets from the two countries without gender data in the analyses but of course there was no outcome of gender statistics for these 2 countries.

OK.

Line 145 mentions 31000 results, line 166 mentions 10302 results? What happened in the selection procedure and why do you call the 31000 results the final database?

On line 183 we mention the participation of 10,302 subjects. As we received from each subject one serum samples in which we measured 3 different vaccine IgG antibody concentrations, we obtained 30,906 results. We mentioned on line 156 approximately 31,000 results.

OK.

Line 170: Intermediate levels high in Norway, vaccination programs?

The possible reasons for these high intermediate levels in Norway are not known. It might be that the vaccination program has had influence on these levels but that seems not so likely due to the older age of our participants. Another explanation might be the higher circulation of Bordetella in Norway considering the high incidence numbers in Norway on the ECDC website for many years now. This might mean that people from these age groups have a higher chance of exposure to the pathogen.

not clear to me why circulation of pertussis would be higher in Norway. Further explanations may be needed.

Line 238: subdivisions of countries and time periods is not discussed in the methods or results section; could you please elaborate in the respective sections to introduce these discussion items

We have added a sentence for this arbitrarily chosen subdivision in the Materials and Methods section on lines 167-170.

dilatory response—stating that a choice is arbitrary does not make it less arbitrary...

Line 272: this refers to the same references as mentioned at the start of your introduction. Maybe be more precise in the introduction about the 500/100000 and not >100

We thank the reviewer for this observation; we have changed this in notification rate up to 50/100,000 according to the ECDC atlas for Pertussis on line 83 and 305.

OK.

Line 294 and following: Since the infection is toxin mediated, protection will most probably be through the antibodies of course. But could it be that another immunological mechanism could play a role, e.g. T cell responses? Susceptibility is very high in some groups yet no major outbreaks occur. An interesting other source could be a link to the European diphtheria surveillance network. They published last year a nice manuscript on Diphtheria. Naresh Chand Sharma, Androulla Efstratiou, Igor Mokrousov, Ankur Mutreja, Bhabatosh Das & Thandavarayan Ramamurthy. Nature Reviews Disease Primers volume 5, Article number: 81 (2019). These authors refer to the gender effect in protection from diphtheria as well as on the low rate of serological immunity among adults depending on the time delay in vaccination during their childhood, that is, a delay up to 3 years.

Of course cellular immunity plays also a very important role in immunity and therefore protection against pertussis by IgG-PT antibody levels is just a part of the whole story. But for obvious reasons we were not able to investigate these kind of immunological parameters in this study.

We thank the reviewer for the reference of the nice publication of the diphtheria network and we have added this observation with the reference on lines 339-340.

very important comment, but unfortunately the authors' reply is rather dilatory. (Incidentally, I understand the reviewer refers to diphtheria immunity, but the authors discuss only pertussis.) As I wrote in my second review, the authors' claim that low levels of anti-PT IgG correlate with susceptibility to pertussis is unfounded.

Reviewer #1 (Remarks to the Author):

Reviewer's report on: "Widespread circulation of pertussis and poor protection against diphtheria among middle-aged adults in 18 European countries"

I thank the authors for their thorough responses. The revised manuscript is now clearer and better emphasizes the caveats that concern serological studies for pertussis. Nevertheless, I still have a number of substantive comments that need to be addressed.

Major comments

1. The authors interpret high interpret high PT antibody titers as evidence of recent infection (or exposure), but it would be useful to clarify what they mean by that exactly and how what is estimated relates to standard epi-demiological quantities (like prevalence or incidence of infection). Tools have been developed (by Peter Teunis and colleagues) and implemented in R (<https://cran.r-project.org/web/packages/seroincidence/index.html>) to estimate sero-incidence rates from cross-sectional data like those in this study. A second step would be to translate those sero-incidence estimates into estimates of sero-prevalence (that is, of the proportion of individuals infectious at a given time point). It is not easy to connect these two quantities in general, but even a rough approximation like $p = \Lambda D$ (where Λ is the sero-incidence rate estimate and D the infectious period of pertussis, approximately 3 weeks) would be useful. To illustrate, if we assume that the proportion with anti-PT IgG ≥ 100 IU/mL estimated in this study (denoted by f) represents infection in the past 6 months [1], then approximately $\Lambda = 2 f$ per year (using the tools of the seroincidence R package will provide a much better estimate). Hence, $p \approx \Lambda D = 2 \times \frac{3}{52} f \approx 0.12 f$, and this simple calculation suggests that the prevalence of infection is an order of magnitude smaller than the fraction estimated in this study.

The reviewer suggests to compare our results with the ones obtained by seroincidence calculations. This is a good suggestion. We have applied the seroincidence tool on our data and this resulted in incidences varying between 0.02 /year (Finland) and 0.05 /year (Norway). These numbers correspond to the ones found in reference 42. The seroprevalence can then simply be estimated by multiplying these incidences with the period of seropositivity. We would like to emphasize that this is certainly not the same as the infectious period, as the reviewer suggests but after personal communication with Peter Teunis he used the period of seropositivity for the infection rate calculation (see also refs 40 and 41). Seropositivity is known to last much longer but varies considerably from person to person. IgG-PT levels above 100 IU/ml typically last 100 to 1000 days of more (3 months to 3 years, ref 41). Multiplying a typical seroincidence rate of, say 0.035 /person-year, by the median seropositivity period (i.e. IgG-PT ≥ 100 IU/ml) of 320 days (ref 41), results in a seroprevalence of 3.1%, perfectly in agreement with our findings. We have added some sentences about this in the discussion on lines 313-317.

2. In their response, the authors cite a paper about recommended cut-offs for serological diagnosis of pertussis [2]. In that paper (Table 2), the IgG anti-PT cutoff of 100 IU/mL is indicated to have $S_e = 78\%$ sensitivity and $S_p = 98\%$ specificity. Using these values to interpret (and adjust) the results of this study would be informative. Specifically, one can use the standard formula relating the true prevalence of infection p_{true} to the estimated prevalence p_{est} : $p_{\text{est}} = S_e p_{\text{true}} + (1 - S_p)(1 - p_{\text{true}})$. Application of this formula for $p_{\text{true}} = 0$ (i.e., no circulation of pertussis) and using the values of S_e and S_p indicated above, one finds that $p_{\text{est}} = 1 - S_p = 0.02$. Under the generous assumption that $f = p_{\text{est}}$ (in reality the estimated prevalence of infection will be much smaller than that, cf. my first comment), this means that estimates below 2% in this study (like in Finland, cf. Table 2) are in fact consistent with no circulation of pertussis at all (!). For values above 2%, one can inverse the formula to adjust the prevalence estimate: $p_{\text{true}} = (p_{\text{est}} + S_e - 1) / (S_e + S_p - 1)$. Applying this formula for p_{est} in the range 4–6.4% (as indicated in the abstract) suggests

that the true prevalence is over-estimated by up to 50% in this study. Again, this is under the generous assumption that $f = p_{est}$; the simple calculation above suggests that p_{est} will actually be much smaller, so that in many countries (not just Finland) it will likely fall into the interval 0–2% associated with no circulation of pertussis (because of the imperfect specificity of serological testing).

Hence, once imperfect sensitivity and specificity are taken into account, the results of this study appear to indicate very little circulation of pertussis and the authors' claim that pertussis circulation is widespread cannot be accepted until further analyses (like those proposed above) are conducted.

*The reviewer has a good point that the sensitivity and specificity should be taken into account. Using the provided estimates of $Se = 0.78$ and $Sp = 0.98$ from the paper of Guiso, Berbers et al. (ref 14), we adjusted the binomial regression model accordingly. It is quite easy to adjust the logit-link in the binomial regression model function to include the Se and Sp . By doing so, the estimates automatically become unbiased and are completely similar to the adjusted prevalence estimates using the standard formula provided above by the reviewer. One advantage of using it directly in the regression model is that we can easily obtain adjusted p -values as well. We have incorporated these new results for $PT \geq 100$ IU/ml in Table 2 and also adjusted Figure 2A and Supplemental Table 1. As the reviewer predicted the proportion of seroprevalence for all countries between 4-6.4% decreased to 2.7-5.8%. Only two countries (Finland and Hungary) fall now in the interval 0-2% associated with no circulation of pertussis and one country (Denmark, 8.4%) remained the same and one country (Norway) actually increased from 9.4 to 9.7%. However, in Finland the reported number of pertussis during the study period (2015 and 2016) was 597 including 31 culture-proven cases, suggesting that *B. pertussis* is actually circulating in this country as well. We adjusted the manuscript with these new data in the Methods, Results and in the Discussion on lines 159-161, 186-187, 193, 234-235 and 278-285, respectively. These new results seem to indicate on (low) circulation in most countries and only in two countries the serosurveillance study indicates to no circulation at all. Surprisingly in these two countries (Finland and Hungary) pertussis case have been notified during 2015-2018 (Supplemental Table 2). In addition, with widespread circulation in the title we meant to emphasize the geographical distribution of pertussis over the participating countries and thus all over Europe and rather not the amount of infections. But to avoid any misunderstandings we have deleted wide-spread from the title of the manuscript.*

Minor comments

The authors write that “[...] antibody concentrations as a marker of waning immunity [...] at least shows one side of the story.” I disagree. As already commented, no serological correlate of protection against pertussis has yet been found. Elsewhere in their replies, the authors comment on the fact that serum antibody levels quickly drop to undetectable levels after vaccination. This is true (see, for example, Ref. [3]), but then one wonders how serum antibodies can be considered as a marker of immunity, given that protection persists even after they have disappeared.

References

- [1] Noel G, Badmasti F, Nikbin VS, Zahraei SM, Madec Y, Tavel D, et al. Transversal sero-epidemiological study of *Bordetella pertussis* in Tehran, Iran. PLoS One. 2020;15(9):e0238398.
- [2] Guiso N, Berbers G, Fry NK, He Q, Riffelmann M, Wirsing von König CH, et al. What to do and what not to do in serological diagnosis of pertussis: recommendations from EU reference laboratories. Eur J Clin Microbiol Infect Dis. 2011 Mar;30(3):307–12.
- [3] Gustafsson L, Hallander HO, Olin P, Reizenstein E, Storsaeter J. A controlled trial of a two-component acellular, a five-component acellular, and a whole-cell pertussis vaccine. N Engl J Med. 1996 Feb;334(6):349–55.

We agree with the reviewer that there is no serological correlate of protection against pertussis, and that protection is still possible by cellular immunity when antibodies have disappeared below detectable levels. In the sentence above we used the term marker as an indication for waning immunity not as a defined biological marker. Therefore we have adjusted this sentence.

Reviewer #1 for reviewer #3 (Remarks to the Author):

Reviewer 1 remarks are in red

Reviewer #3 (Remarks to the Author):

This is a very nice international seroprevalence study on Diphtheria, pertussis and tetanus. Sometimes details are missing for a better understanding of the methodology.

A few general remarks:

Did you correlate the titers between the different antibodies? for example did the unprotected for diphtheria also have lower tetanus levels?

We did not correlate the concentrations of the different vaccine antibodies with each other because we knew from earlier studies that these levels do not correlate. Here we show the 3 correlation graphs for 7 countries which clearly illustrate that there is no real correlation between these antibody levels. For some participants a unprotected diphtheria antibody level correlates with a very low tetanus level but for the majority this is not the case. We have added a sentence at the end of the discussion emphasizing this on line 354-355.

OK. The graphs indeed show no correlation, but it may nevertheless be interesting to add those in the supplement.

We believe that the addition of these graphs does not contribute to the main message of the manuscript and therefore we choose to mention this in the discussion but not to include these graphs.

Line 135: What is meant by recent?

For this arbitrarily chosen cut-off of 100 IU/mL recent means mostly an pertussis exposure in the last

year but sometimes it is also used for 0.5 year. Therefore we like to use the term recent.

This is a critical comment, which is not properly addressed in my opinion. In my second review, I proposed further analyses to connect the proportions estimated in this study to more meaningful epidemiological quantities like prevalence or incidence rate. Doing so would considerably strengthen the paper and facilitate the interpretation of the results.

See above for our answers on this comment. We have performed the proposed further analyses and incorporated the new results in the manuscript.

Line 170: Intermediate levels high in Norway, vaccination programs?

The possible reasons for these high intermediate levels in Norway are not known. It might be that the vaccination program has had influence on these levels but that seems not so likely due to the older age of our participants. Another explanation might be the higher circulation of Bordetella in Norway considering the high incidence numbers in Norway on the ECDC website for many years now. This might mean that people from these age groups have a higher chance of exposure to the pathogen.

not clear to me why circulation of pertussis would be higher in Norway. Further explanations may be needed.

We have not included this in the discussion because we have real explanation for the high circulation of Bordetella in Norway. However, the reported number of pertussis cases in Norway has been higher during last two decades and might be reflecting a very good national notification system. It is also known that the number of samples tested at routine diagnostic laboratories in Norway is high.

Line 238: subdivisions of countries and time periods is not discussed in the methods or results section; could you please elaborate in the respective sections to introduce these discussion items

We have added a sentence for this arbitrarily chosen subdivision in the Materials and Methods section on lines 167-170.

dilatory response—stating that a choice is arbitrary does not make it less arbitrary...

We have chosen these the subdivision of countries and time periods on arbitrarily grounds as described in the Methods

Line 294 and following: Since the infection is toxin mediated, protection will most probably be through the antibodies of course. But could it be that another immunological mechanism could play a role, e.g. T cell responses? Susceptibility is very high in some groups yet no major outbreaks occur. An interesting other source could be a link to the European diphtheria surveillance network. They published last year a nice manuscript on Diphtheria. Naresh Chand Sharma, Androulla Efstratiou, Igor Mokrousov, Ankur Mutreja, Bhabatosh Das & Thandavarayan Ramamurthy. Nature Reviews Disease Primers volume 5, Article number: 81 (2019). These authors refer to the gender effect in protection from diphtheria as well as on the low rate of serological immunity among adults depending on the time delay in vaccination during their childhood, that is, a delay up to 3 years.

Of course cellular immunity plays also a very important role in immunity and therefore protection against pertussis by IgG-PT antibody levels is just a part of the whole story. But for obvious reasons we were not able to investigate these kind of immunological parameters in this study.

We thank the reviewer for the reference of the nice publication of the diphtheria network and we have added this observation with the reference on lines 339-340.

very important comment, but unfortunately the authors' reply is rather dilatory. (Incidentally, I understand the reviewer refers to diphtheria immunity, but the authors discuss only pertussis.) As I wrote in my second review, the authors' claim that low levels of anti-PT IgG correlate with susceptibility to pertussis is unfounded.

We agree that we accidently answered on this comment for IgG-PT but of course the same argument is valid for Diphtheria antibody immunity. We had discussed these possibilities for pertussis on lines 289-291. And we have added some sentences about cellular immunity for diphtheria on lines 339-341.

REVIEWERS' COMMENTS

Reviewer #1 (Remarks to the Author):

I thank the authors for their detailed responses and additional analyses. I have no further comments.